# Tooth replacement patterns in the Early Triassic epicynodont *Galesaurus planiceps* (Therapsida, Cynodontia)

Luke A. Norton[1,2]*, Fernando Abdala[1,3], Bruce S. Rubidge[1,2], Jennifer Botha[4,5]

**1** Evolutionary Studies Institute, University of the Witwatersrand, Johannesburg, Gauteng, South Africa, **2** School of Geosciences, University of the Witwatersrand, Johannesburg, Gauteng, South Africa, **3** Unidad Ejecutora Lillo (CONICET-Fundación Miguel Lillo), Tucumán, Argentina, **4** Karoo Palaeontology, National Museum, Bloemfontein, Free State, South Africa, **5** Department of Zoology and Entomology, University of the Free State, Bloemfontein, Free State, South Africa

* luke.norton@students.wits.ac.za

**Data Availability Statement:** All relevant data are within the paper and its Supporting information files.

## Abstract

Sixteen specimens of the Early Triassic cynodont *Galesaurus planiceps* (including eight that were scanned using micro-computed tomography) representing different ontogenetic stages were assembled to study the dental replacement in the species. The growth series shows that the incisors and postcanines continue to develop and replace, even in the largest (presumably oldest) specimen. In contrast, replacement of the canines ceased with the attainment of skeletal maturity, at a basal skull length of ~90 mm, suggesting that *Galesaurus* had a finite number of canine replacement cycles. Additionally, the functional canine root morphology of these larger specimens showed a tendency to be open-rooted, a condition not previously reported in Mesozoic theriodonts. An alternating pattern of tooth replacement was documented in the maxillary and mandibular postcanine series. Both postcanine series increased in tooth number as the skull lengthened, with the mandibular postcanine series containing more teeth than the maxillary series. In the maxilla, the first postcanine is consistently the smallest tooth, showing a proportional reduction in size as skull length increased. The longer retention of a tooth in this first locus is a key difference between *Galesaurus* and *Thrinaxodon*, in which the mesial-most postcanines are lost after replacement. This difference has contributed to the lengthening of the postcanine series in *Galesaurus*, as teeth continued to be added to the distal end of the tooth row through ontogeny. Overall, there are considerable differences between *Galesaurus* and *Thrinaxodon* relating to the replacement and development of their teeth.

## Introduction

*Galesaurus planiceps* [1] is a small non-mammaliaform cynodont with a maximum known skull length of 114 mm. It was the first cynodont from South Africa to be described, and evidence of the heterodonty in this taxon later led to the establishment of the Cynodontia in order to differentiate the genus from other fossil 'reptiles' recovered from the region [2].

**Funding:** This research was funded by the National Research Foundation (www.nrf.ac.za) Award ID UID 95980 to JB, and a Professional Development Programme Doctoral Scholarship to LAN; the Palaeontological Scientific Trust (www.past.org.za) to JB and BSR; the DSI/NRF Centre of Excellence in Palaeosciences (www.wits.co.za/coepalaeo) to JB and BSR; and the Consejo Nacional de Investigaciones Científicas y Técnicas (www.conicet.gov.ar) to FA. The funders had no role in study design, data collection and analysis, decision to publish, or preparation of the manuscript.

**Competing interests:** The authors have declared that no competing interests exist.

Remains of *Galesaurus* are known exclusively from the Lower Triassic *Lystrosaurus declivis* Assemblage Zone of the South African Main Karoo Basin [3]. The first appearance of the genus in the stratigraphic record is approximately 22 m above the Permian–Triassic Boundary (PTB), in the upper Palingkloof Member of the Balfour Formation, Beaufort Group [4, 5]. The appearance of the taxon close to the PTB suggests that *Galesaurus*, or at least a ghost lineage of the Galesauridae, survived the end-Permian mass extinction [6–9].

*Galesaurus* occurred contemporaneously with the closely allied taxon, *Thrinaxodon liorhinus* [4, 8], but its stratigraphic range is more constrained [10], with the last occurrence of *Galesaurus* approximately 85 m above the PTB within the lower Katberg Formation [4, 5]. In contrast, the range of *Thrinaxodon* extends to the top of the Katberg Formation [5]. Despite its short stratigraphic range, *Galesaurus* is the second most abundant cynodont (after *Thrinaxodon*) recovered from the *Lystrosaurus declivis* Assemblage Zone [11, 12], with over 30 specimens attributed to the genus [13].

*Galesaurus* has a cranial morphology generally reminiscent of *Thrinaxodon* but, as in the more basal *Procynosuchus*, the osseous secondary palate in *Galesaurus* has a wide cleft between the maxillary and palatine processes [14, 15]. In contrast, the cleft in *Thrinaxodon* is relatively narrow with the palatal plates nearly in contact [16]. Interestingly, *Galesaurus* also presents characters only recognized in some members of the more derived Eucynodontia [17]. These include an angulation between the ventral edge of the maxillary zygomatic process and the anteroventral margin of the jugal, and a well-projected posterodorsal lamina of the zygomatic portion of the squamosal [18].

Despite the close morphological and hypothesized behavioral [19–22] affinities between *Galesaurus* and *Thrinaxodon* (which even resulted in the first specimens of *Thrinaxodon* to originally be identified as *Galesaurus* [23–25]), the former has received comparatively little research interest. This is likely due to the fact that *Thrinaxodon* has been traditionally considered as a hypothetical model ancestor to the Mammalia [15, 26, 27]. An important feature for this key placement was that the postcanine morphology of *Thrinaxodon* resembles that of the early mammal *Morganucodon* [28–30] (= Mammaliaformes [31–38]). The relative abundance of *Thrinaxodon* has indeed also contributed to its detailed study, which includes a large body of literature focusing on the tooth replacement pattern [39–44]. Only the Middle Triassic gomphodont cynodont *Diademodon* has undergone a comparable amount of research on tooth replacement [45–49]. In contrast, only recently has a study reporting tooth replacement of the maxillary canines in a single specimen of *Galesaurus* been published [50]. Determining the tooth replacement patterns in *Galesaurus* and comparing them to those observed in the previously studied cynodont *Thrinaxodon* [44] may help to ascertain whether any variation in tooth replacement pattern occurred amongst the basal-most cynodonts.

Owen [1] described the postcanine dentition of *Galesaurus* (Fig 1) as being simple-crowned and of equal size. Van Hoepen [51] later described a new taxon, *Glochinodon detinens*, as having a morphologically unique postcanine dentition, where the teeth are bicuspid, with a large anterior cone, reflected posteriorly over a smaller posterior cusp. Watson [52] redescribed the holotype of *Galesaurus*. Watson's account of the dentition mostly agreed with Owen's original description, but he added that the postcanines were laterally compressed, such that they are oval in cross-section, with the maximum length being in the mesiodistal direction. Watson [52] also noted that there is no evidence of tooth replacement in the holotype. Broom [53] later examined the material of *Galesaurus* and *Glochinodon* and noticed a similar bicuspid morphology in impressions of postcanine crowns of *Galesaurus*. This shared postcanine morphology led to the synonymisation of the two genera [53, 54]. Parrington [55] also provided a brief description of the dentition of *Galesaurus*, but did not add much more detail regarding the postcanine dentition. He concluded that the maxillary postcanine series contained fewer

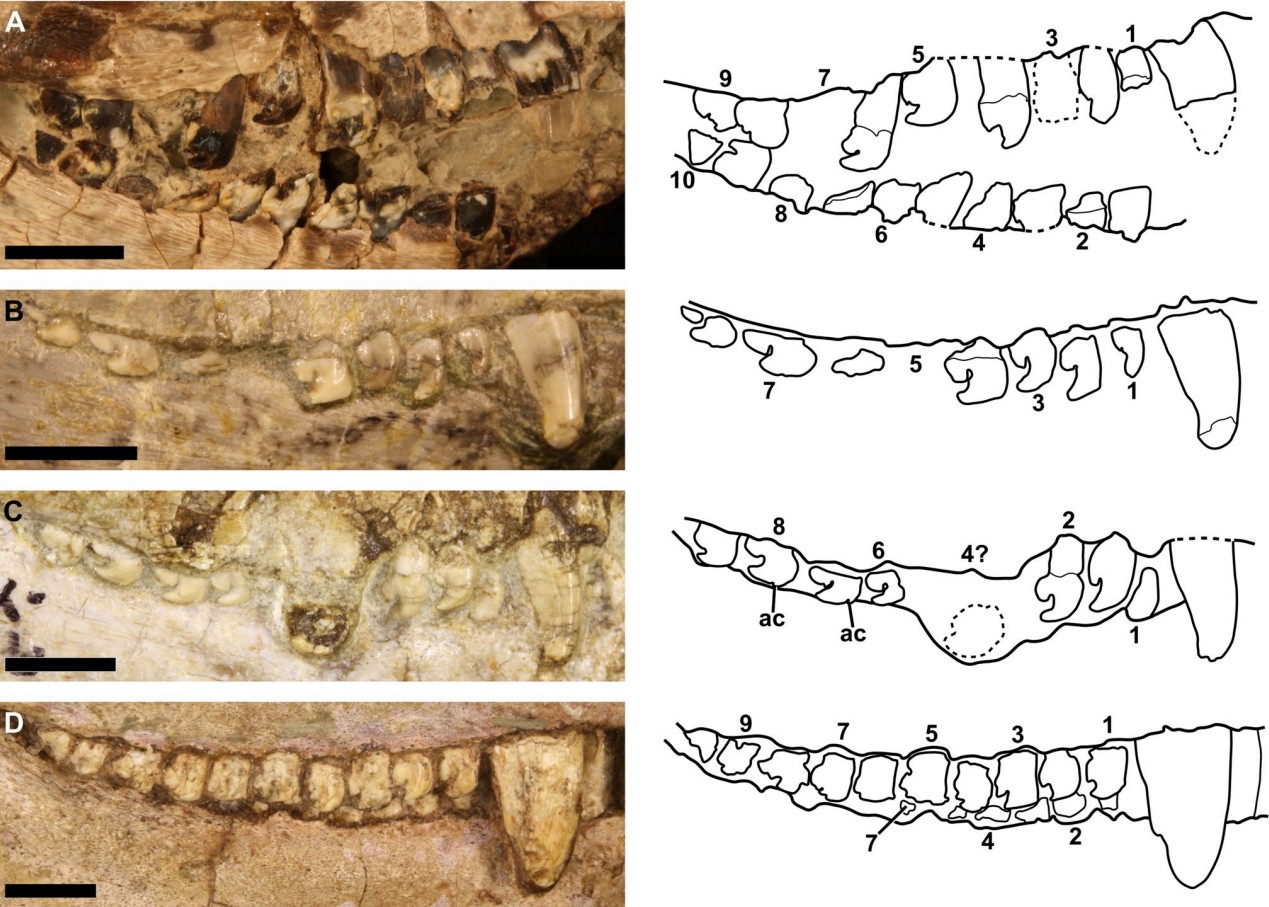

**Fig 1. Postcanine morphology of *Galesaurus planiceps*.** (A) Photograph and interpretive drawing of the left maxillary and mandibular dentition of a juvenile (NMP 581, mirrored for comparative purposes). (B) Photograph and interpretive drawing of the right maxillary dentition of a subadult (RC 845). (C) Photograph and interpretive drawing of the right maxillary dentition of a subadult (SAM-PK-K9956). (D) Photograph and interpretive drawing of the right maxillary dentition of an adult (NMQR 3340). Abbreviation: ac, accessory cusp. Arabic numerals indicate postcanine positions. Scale bars equal 5 mm.

teeth than the mandibular series, contradicting Broom's [54] notion that the two series had an equal number of elements. Parrington [55] also documented the presence of replacement incisors in the dentaries.

*Glochinodontoides gracilis* [56] was described as having postcanine teeth of the same peculiar morphology, but smaller and with a shorter postcanine series than that of *Glochinodon*. A second specimen attributed to *Glochinodontoides* was described as having postcanine teeth with a less curved anterior cusp than *Galesaurus* [57]. *Glochinodon* and *Glochinodontoides* were later considered to be synonymous with *Galesaurus* [53, 54, 57, 58]. Additionally, it was noted by Hopson and Kitching [58] (pp. 73–74) that "small, immature specimens have usually been referred to the genus *Galesaurus*, [and] large, mature specimens to *Glochinodontoides*."

Due to increased effort in the collection of fossils stratigraphically close to the PTB (e.g., fieldwork undertaken by R.M.H. Smith and J. Botha), the number of *Galesaurus* specimens in museum collections has increased during the last 20 years, generating a multitude of studies inspecting the taxon from different perspectives. The most recent significant research on the biology of *Galesaurus*, have been studies of the postcranial skeleton and paleohistology [20, 59], ontogeny of the cranium [13], evidence for parental care [22], and the tooth attachment

system [60]. In addition, a detailed description of the internal cranial anatomy of a single sub-adult specimen of *Galesaurus* (AMNH FARB 2227) has recently been published [50].

As the mandible is preserved in tight occlusion in most specimens of *Galesaurus*, in the past only the labial surfaces of the upper dentition were accessible for study. Recent advancements in three-dimensional (3-D) scanning techniques have allowed studies on the dentition of therapsid fossils to be undertaken *in silico* [44, 50, 60–62], negating the need to permanently damage material to study internal morphologies, as was common practice in the past (e.g., [41, 63–66]). By μCT-scanning selected specimens, it was possible to describe the entire dental complement of *Galesaurus* for the first time. The μCT data also allowed for *in silico* observation of tooth replacement patterns as unerupted replacement teeth, functional teeth, and partially resorbed roots of shed teeth were easily distinguishable.

This paper presents a detailed analysis of the tooth replacement of *Galesaurus*, and a comparison with the tooth replacement patterns of *Thrinaxodon liorhinus* from previous studies [39, 40, 42–44]. Micro-computed tomography scanning of an ontogenetic series facilitated a comprehensive examination of the tooth replacement in upper and lower dental series comprising specimens of different sizes. This study provides the first comparison of tooth replacement patterns between two contemporaneous non-mammaliaform cynodonts. This assessment is of particular interest as it offers a comparison of biological processes (e.g., tooth replacement), as opposed to a simple comparison of anatomical and morphological characters of the two genera.

### Institutional abbreviations

AMNH, American Museum of Natural History, New York, USA; BP, Evolutionary Studies Institute (formerly Bernard Price Institute for Palaeontological Research), University of the Witwatersrand, Johannesburg, South Africa; FMNH, Field Museum of Natural History, Chicago, USA; NMP, Natal Museum, Pietermaritzburg, South Africa; NMQR, National Museum, Bloemfontein, South Africa; RC, Rubidge Collection, Wellwood, Graaff-Reinet, South Africa; SAM, Iziko: South African Museum, Cape Town, South Africa; TM, Ditsong National Museum of Natural History (formerly Transvaal Museum), Pretoria, South Africa.

## Materials and methods

There are currently more than 30 specimens attributed to *Galesaurus planiceps* [13]. Seventeen of these specimens are sufficiently prepared such that the dentition may be studied (Table 1). These specimens represent a presumed ontogenetic growth series based on the basal skull length (BSL), which ranges from 62 mm (FMNH PR 1774) to 114 mm (NMQR 860).

### Micro-computed tomography scanned specimens

Eight specimens of *Galesaurus planiceps*, representing an ontogenetic series consisting of sub-adult (BSL 69–88 mm) and adult (BSL > 90 mm) specimens [13], were analyzed using μCT (Table 2). The BSL of this sample ranges from 69 mm (RC 845) to 114 mm (NMQR 860). All specimens were scanned using a Nikon Metrology XTH 225/320 LC dual source industrial CT system at the Wits Microfocus X-ray CT Facility (Evolutionary Studies Institute, University of the Witwatersrand, Johannesburg, South Africa).

Due to differing physical dimensions, states of preparation, and chemical compositions of the specimens, the scan parameters were adjusted to obtain the best results for each specimen at 4000 projections (Table 2). In order to reduce beam-hardening artifacts, a 1.2 mm copper or 1.8 mm aluminum filter was used [68]. The resulting scans ranged in isotropic voxel sizes from 42.6 μm (RC 845) to 80 μm (NMQR 3542).

**Table 1. Specimens of *Galesaurus planiceps* included in this study, listed in increasing size.**

| Specimen | BSL (mm) | Ontogenetic stage | Postcanine count (L/R) | |
|---|---|---|---|---|
| | | | Maxilla | Mandible |
| FMNH PR 1774[a] | 62 | Juvenile | 7/7 | 9/9 |
| NMP 581 | 64 | Juvenile | 9/9 | 10?/– |
| RC 845[b] | 69 | Subadult | 10/9 | 11/11 |
| SAM-PK-K1119 | 72 | Subadult | 8/9 | 9/? |
| SAM-PK-K9956 | 73 | Subadult | –/9 | – |
| NMQR 655 | ~75 | Subadult | 8/? | – |
| AMNH FARB 2227[c] | 79 | Subadult | 10/10 | ?/11 |
| BP/1/4714[b] | 81 | Subadult | 11/11? | 12/13 |
| BP/1/4602[b] | 88 | Subadult | 10/10 | 11/11 |
| BP/1/3892[d] | 90 | Adult | 9/9 | – |
| NMQR 1451 | 90 | Adult | 10/10? | – |
| NMQR 135[b] | 94 | Adult | 9/9? | – |
| NMQR 3340 | ~102 | Adult | 10/10 | – |
| NMQR 3542[b] | 102 | Adult | 11/11 | 15?/12 |
| BP/1/5064[b] | 103 | Adult | 11/11 | 14/14 |
| SAM-PK-K10468[b] | 105 | Adult | 9/10 | 11/10 |
| NMQR 860[b] | 114 | Adult | 10/10 | 13/13 |

Ontogenetic stages based on Jasinoski and Abdala [13]. Abbreviations: BSL, basal skull length; L, left; R, right. A dash (–) indicates that the maxilla/mandible is not preserved or that the mandible is preserved in tight occlusion; a question mark (?) represents an uncertain count due to partial damage/obstruction of the maxilla/hemimandible.

[a] Specimen serially sectioned, information from Rigney [63]

[b] Specimen μCT-scanned for this study

[c] Specimen μCT-scanned by Pusch et al. [50]

[d] Specimen missing from collection, information from Brink [67]

Three-dimensional rendering and segmentation of the data were performed using VGStudio MAX 2.2.5 (Volume Graphics, Heidelberg, Germany). For all specimens, teeth were segmented as separate structures from the surrounding bone and matrix. Segmentation was undertaken using the semiautomatic 3-D 'region growing' tool. Replacement teeth were recognizable due to the presence of mineralized tissue and are therefore considered as being at least in an advanced bell stage of tooth development [69]. It is during middle–late bell stages that tooth crown morphology is determined due to apposition and mineralization of dentin and enamel [70]. Due to low contrast between tooth and bone in some specimens (e.g., NMQR

**Table 2. Parameters used for μCT-scanning of *Galesaurus planiceps* specimens.**

| Specimen | Voxel size (μm) | Tube voltage (kV) | Tube current (μA) | Frame rate (fps) |
|---|---|---|---|---|
| RC 845 | 42.60 | 140 | 250 | 2 |
| BP/1/4714 | 68.56 | 115 | 135 | 0.5 |
| BP/1/4602 | 66.72 | 130 | 185 | 1 |
| NMQR 135 | 58.80 | 190 | 210 | 1 |
| NMQR 3542 | 80.00 | 110 | 150 | 1 |
| BP/1/5064 | 66.04 | 170 | 95 | 1 |
| SAM-PK-K10468 | 71.30 | 120 | 120 | 1 |
| NMQR 860 | 74.10 | 210 | 300 | 1 |

3542 and NMQR 860), this tool could not be used, and segmentation of the teeth was completed manually using the 'polygon lasso' and 'polyline' tools.

## Normalization of basal skull length measurements

The BSL measurements for *Galesaurus* and *Thrinaxodon* range from 62–114 mm and ~30–96 mm respectively (S1 Dataset). In order to compare changes in the number of postcanines with increased BSL between the two taxa, the measurements were recalculated with min-max feature scaling/unity-based normalization, using the following equation:

$$x' = \frac{BSL_i - BSL_{\min}}{BSL_{\max} - BSL_{\min}} \tag{1}$$

In the equation, $BSL_i$ represents the actual specimen measurement, $BSL_{\min}$ represents the smallest and $BSL_{\max}$ the largest measured BSL values in the range. Feature scaling was chosen as it allows for the recalculation of the values, such that the data occupy ranges from zero to one, without affecting the mean and standard deviation of the sample.

## Results

### Tooth numbers and morphology

**Incisors.** *Galesaurus planiceps* has four upper and three lower incisors. Upper incisor morphology is simple with a distally directed cusp. Both upper and lower incisors usually have a broad base and taper gradually towards the crown apex. In some instances, the mandibular incisors also show tapering towards the root apex. Mandibular incisors tend to have a slight procumbent orientation (e.g., BP/1/4602 and BP/1/5064), and the third mandibular incisor (i3) has a straighter, more dorsal orientation (e.g., RC 845, Fig 2C and 2D).

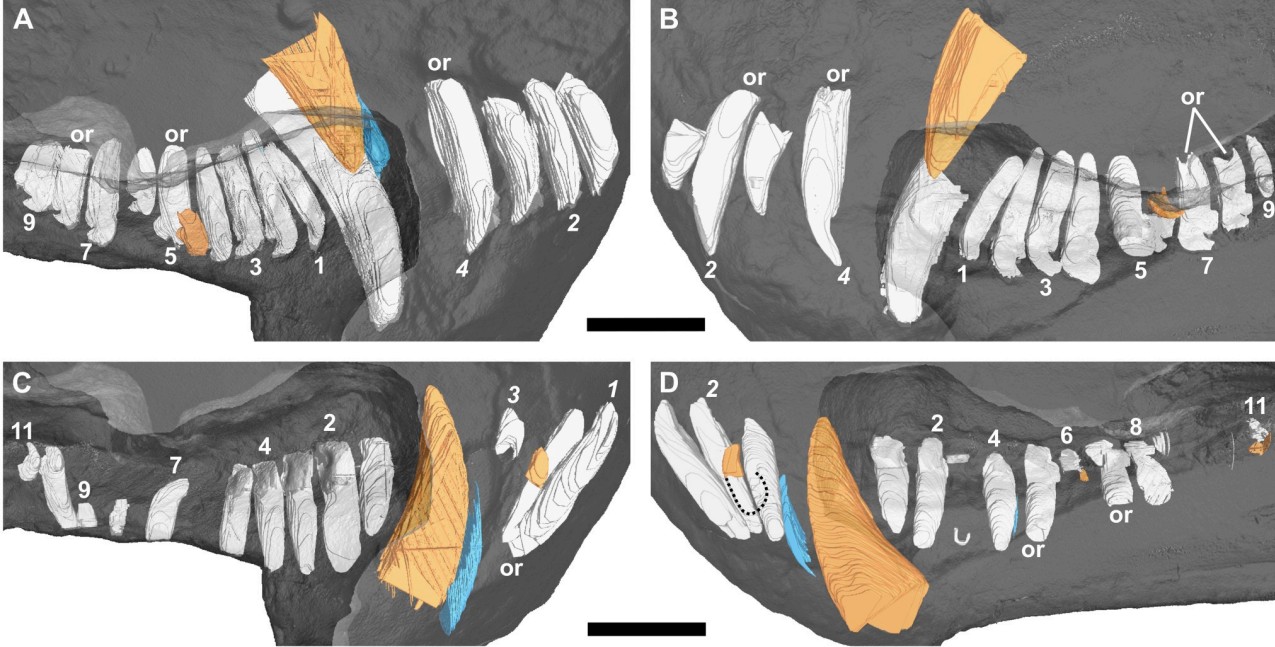

**Fig 2. Three-dimensional rendering of the tooth rows of a subadult *Galesaurus planiceps* (RC 845) in medial view.** (A) Upper left. (B) Upper right. (C) Lower left. (D) Lower right. Note the more procumbent orientation of i1 and i2, in comparison to the straighter orientation of i3 (C and D). Replacement teeth/crypts in orange, old remnant roots in blue. Resorption of the right functional i2 and i3 indicated with a dashed line (D). Abbreviation: or, open root. Arabic numerals indicate incisor (italicized) and postcanine positions. Scale bar equals 10 mm.

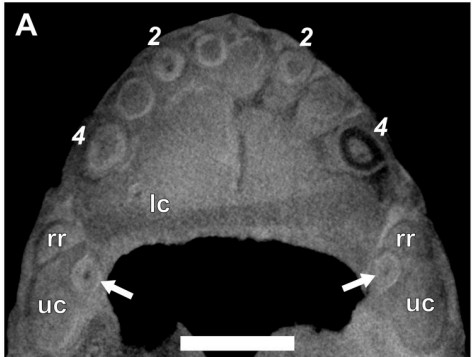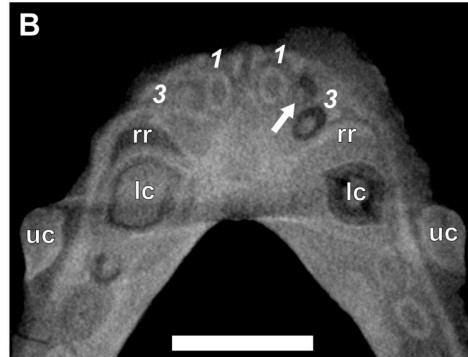

**Fig 3. Virtual horizontal sections through the anterior dentition of a subadult *Galesaurus planiceps* (RC 845).** (A) Premaxillary incisors and maxillary canines. (B) Mandibular incisors and canines. Abbreviations: lc, mandibular canine; rr, remnant root; uc, maxillary canine. Arrows indicate replacement canines and incisors. Arabic numerals (italicized) indicate incisor positions. Scale bars equal 5 mm.

**Canines.**   The canine is a single functional tooth, with a broad, conical morphology. The crowns of both the upper and lower canines are similar in shape, although the latter are slightly more slender. The orientation of the lower canines is almost vertical, whereas the uppers have a slight rostral inclination. The µCT-scanned specimens showed both maxillary (e.g., RC 845, Fig 3A and NMQR 135, Fig 4A) and mandibular canines (e.g., RC 845, Fig 3B and BP/1/4602, Fig 5B) to have prominent lateral ridges.

**Postcanines.**   Postcanine morphology of *Galesaurus* is unique in the Epicynodontia. The teeth of both the upper and lower jaw are mesiodistally elongated and bear two cusps (Fig 6). Crown morphology of the upper and lower postcanines is similar (Figs 1A and 6B–6D). The mesial cusp is larger and curves distally over the smaller distal cusp. Specimens previously attributed to "*Glochinodontoides*" have been described as having a mesial cusp that is longer,

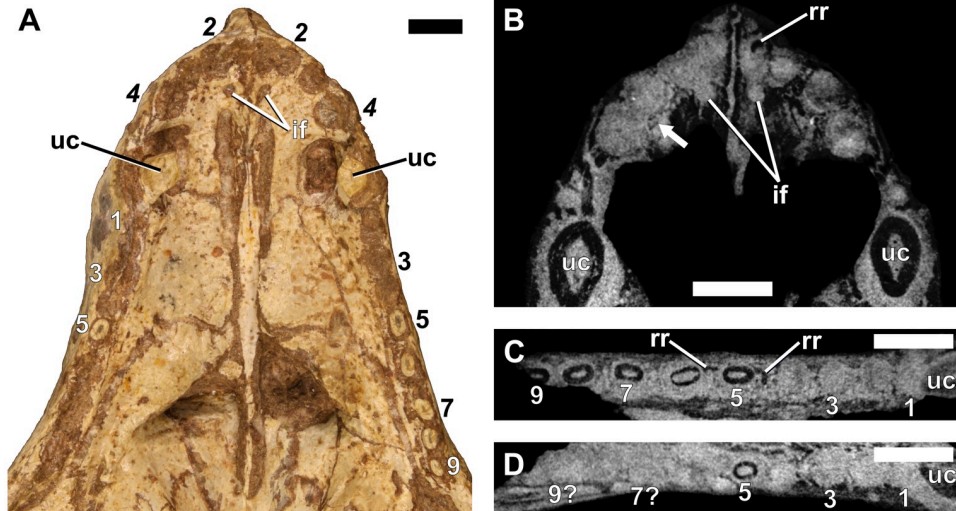

**Fig 4. Dentition of an adult *Galesaurus planiceps* (NMQR 135).** (A) Ventral view of the palate. (B) Virtual horizontal section through premaxilla and anterior maxilla. (C) Virtual horizontal section through left maxillary postcanine series. (D) Virtual horizontal section through right maxillary postcanine series. Abbreviations: if, incisive foramen; rr, remnant root; uc, maxillary canine. Arrow indicates the crypt of a replacement incisor. Arabic numerals indicate incisor (italicized) and postcanine positions. Scale bars equal 5 mm.

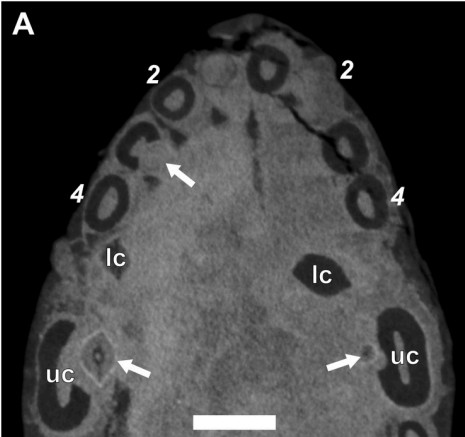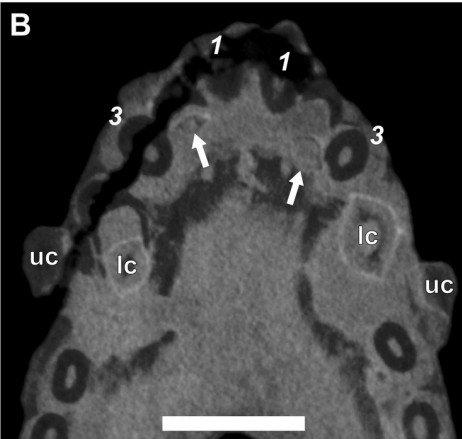

**Fig 5. Virtual horizontal sections through the anterior dentition of a subadult *Galesaurus planiceps* (BP/1/4602).** (A) Premaxillary incisors and maxillary canines. (B) Mandibular incisors and canines. Abbreviations: lc, mandibular canine; uc, maxillary canine. Arrows indicate replacement canines and incisors. Arabic numerals (italicized) indicate incisor positions. Scale bars equal 5 mm.

and not as distally curved as that of *Galesaurus* [57]. In some specimens (e.g., BP/1/4602), the entire postcanine series has elongated mesial cusps, whereas in smaller specimens (BSL < 73 mm, Fig 1A–1C) only the mesial-most postcanines (PC1–PC3) have tall, elongated mesial cusps and the distal teeth have shorter, more recurved mesial cusps (Fig 6D). Jasinoski and Abdala [13] noted the presence of a small accessory cusp mesial to the main recurved cusp in the maxillary postcanines of two small specimens: BP/1/4597 (BSL ~70 mm) and NMQR 3716 (BSL 75 mm). Such accessory cusps were also observed in PC7 and PC8 of the similarly sized SAM-PK-K9956 (BSL 73 mm, Fig 1C). The distal cusp of the first maxillary postcanine is reduced in size in several smaller specimens (e.g., RC 845 and BP/1/4602) such that it is almost indistinguishable. In some larger specimens (e.g., NMQR 3340, Fig 1D), however, the first erupted postcanine has the typical bicuspid crown morphology. The number of postcanines recorded in the study sample varies from seven to 12 in the maxilla, and nine to 15 in the dentary. There do not appear to be any noticeable differences between the crown morphologies of the upper and lower postcanines within an individual. Both the maxillary and mandibular postcanine series are arranged in a slight imbricate pattern such that the mesial portion of the tooth crown is directed lingually, whereas the distal margin of the crown is directed labially.

## Tooth replacement

Evidence of tooth replacement was recorded in all μCT-scanned specimens of *Galesaurus planiceps*. Several specimens facilitated an evaluation of replacement activity for the full dental complement of the same individual. Replacement teeth for the incisors and postcanines are situated lingual to the functional tooth. Replacement canines are usually not erupted. The following section describes the state of replacement of the teeth (i.e., presence of replacement teeth, developmental condition of roots, etc.) for the eight μCT-scanned specimens of *Galesaurus*. These descriptions are presented in order from smallest to largest basal skull length (BSL).

### RC 845 (BSL 69 mm)

**Incisors.** There is no evidence of replacement teeth amongst the maxillary incisors (Fig 3A). On both sides, I4 is the largest tooth and I2 the second largest. On the left (Fig 2A), I1 and

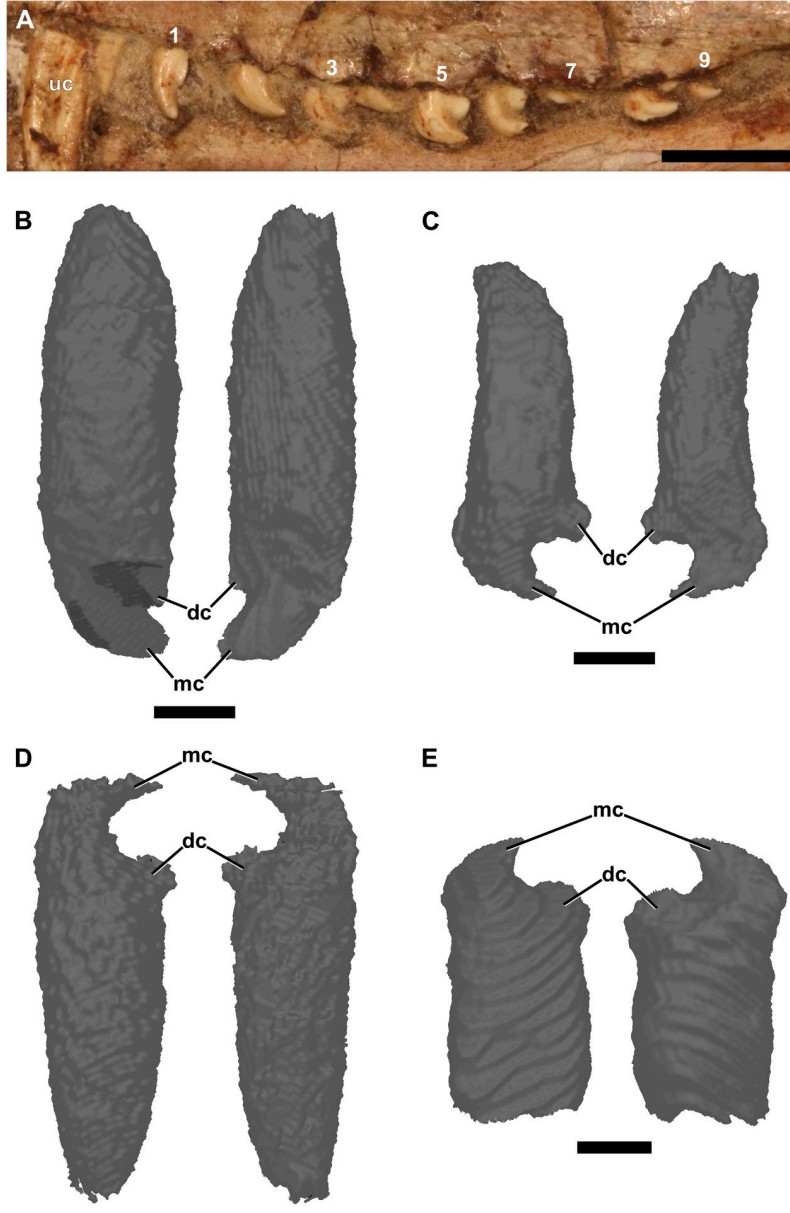

**Fig 6. Three-dimensional renderings of the postcanine teeth of a subadult *Galesaurus planiceps* (BP/1/4602).** (A) Photograph of the left maxillary dentition. (B) Labial and lingual views of the third left maxillary postcanine (PC3). (C) Labial and lingual views of the tenth left maxillary postcanine (PC10). (D) Labial and lingual views of the second left mandibular postcanine (pc2). (E) Labial and lingual views of the eleventh right mandibular postcanine (pc11, mirrored for comparative purposes). Abbreviations: dc, distal cusp; mc, mesial cusp; uc, maxillary canine. Scale bars equal 5 mm (A) and 2 mm (B–E).

I3 are similar in size, but on the right (Fig 2B), I1 is smaller than the I3, both of which have open roots and were still in the process of developing. Replacement teeth of the mandibular i2 are visible on both sides (Fig 2C and 2D), with noticeable resorption of the tooth roots of the functional i2 and i3 visible in the right (Fig 2D). There is no evidence of replacement for the i1

and i3 loci in the mandibular rami. The left i3 is open-rooted, whereas the root of the right i3 is fully developed and tapers to a closed point at the root apex.

**Canines.** A large replacement canine is present anteromedial to the functional canine in both maxillae (Fig 2A and 2B). A remnant canine root is retained in both maxillae, labial and mesial to the functional canine, and it is evident that the replacement canines have slightly eroded the roots of the functional teeth (Fig 3A). The lower canines were still in the process of erupting, with the left being slightly more developed (Fig 2C and 2D). The newly erupted replacement mandibular canines do not appear to have associated replacement teeth. Mesial to the newly erupting functional canine there appear to be partially resorbed roots of remnant teeth present in both rami (Figs 2C, 2D and 3B).

**Postcanines.** Ten postcanine teeth are present in the left maxillary series (Figs 2A and 7A). In both maxillae, the root of PC1 is closed and does not make contact with the functional canine. Open roots are observed for PC5 and PC8. A partially erupted tooth is preserved between the crowns of PC4 and PC5. This tooth is interpreted as a developing replacement based on its partially mineralized root and lingual position to the functional tooth row. Furthermore, this replacement is identified as the emerging PC4 due to it being located distal to the fourth locus. In the right maxillary series (Figs 2B and 7B), nine postcanine teeth are preserved. The root of PC6 has been resorbed, and a replacement crown was in the process of forming dorsolingually to the functional tooth. Open roots are present in PC7 and PC8.

In the left mandibular series (Figs 2C and 7C), 11 alveoli and 10 postcanines are preserved (pc6 is absent), and there is no evidence of postcanine replacement. The ventral part of the hemimandible, distal to pc4, is lost such that the root apices are not preserved. It is assumed that this damage had no impact on the preservation of any replacement postcanines in the left dentary, as the replacement teeth in the right hemimandible (Fig 2D) are situated more dorsal in the dentary corpus than the region missing from the left. In the right mandibular series (Figs 2D and 7D) 11 postcanine teeth are present. The crowns of pc3 and pc6 are present, despite the almost complete resorption of the roots, whereas the roots of pc5 and pc7 are still open. An old root is preserved distal to pc4. There is no evidence for mineralized replacement teeth, however, crypts are associated with pc6 and pc11.

## BP/1/4714 (BSL 81 mm)

**Incisors.** In the left premaxilla, only the I1 and I4 are preserved, with no evidence of replacement teeth (Fig 8A). Only I3 and I4 of the right series are present (Fig 8B). Replacement

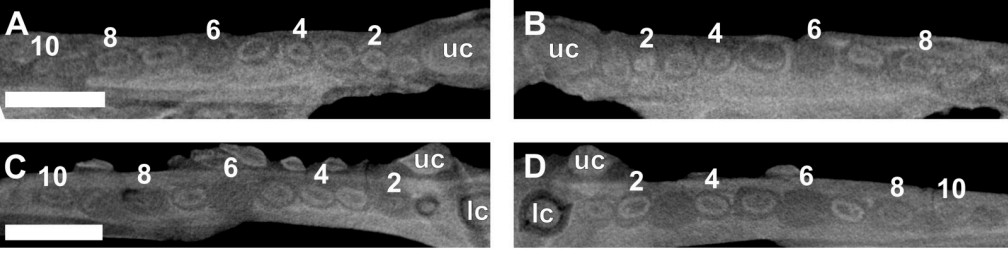

**Fig 7. Virtual horizontal sections through the maxillae and dentaries of a subadult *Galesaurus planiceps* (RC 845).** (A) Left maxillary postcanine series. (B) Right maxillary postcanine series. (C) Left mandibular postcanine series. (D) Right mandibular postcanine series. Abbreviations: lc, mandibular canine; uc, maxillary canines. Arabic numerals indicate postcanine positions. Scale bars equal 5 mm.

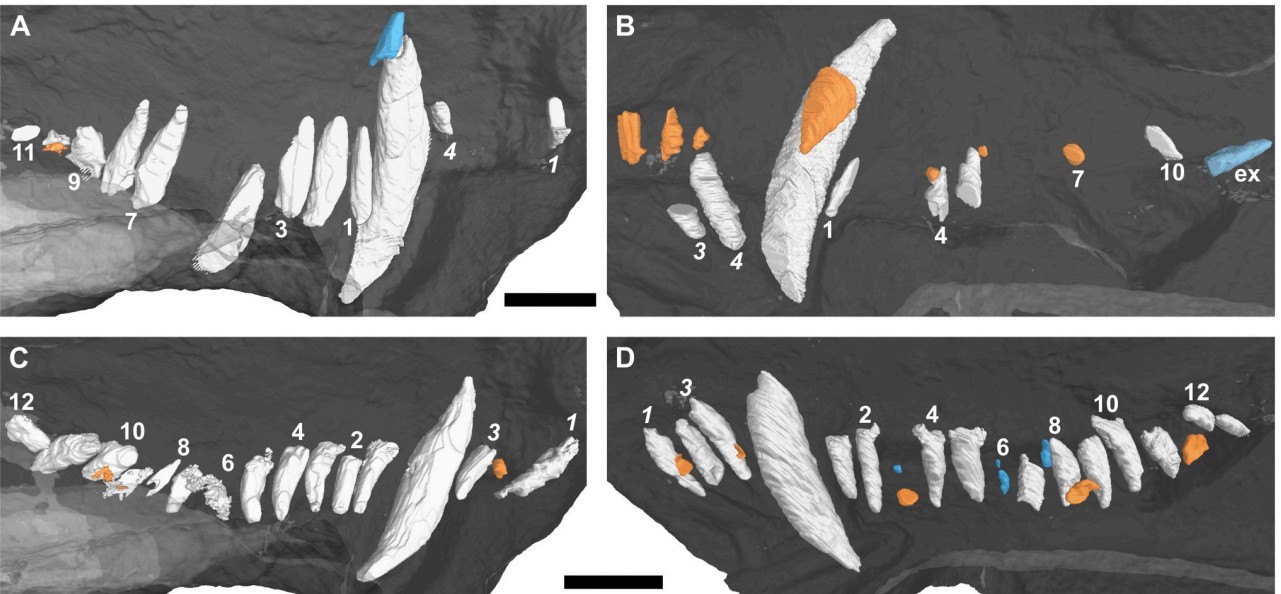

**Fig 8. Three-dimensional rendering of the tooth rows of a subadult *Galesaurus planiceps* (BP/1/4714) in medial view.** (A) Upper left. (B) Upper right. (C) Lower left. (D) Lower right. Replacement teeth in orange, old remnant roots in blue. Abbreviation: ex, *ex situ* postcanine. Arabic numerals indicate incisor (italicized) and postcanine positions. Scale bars equal 20 mm.

teeth are present lingually for the second to fourth incisors. The replacement element of I4 is remarkably smaller compared to those of the mesial incisors. Only i1 and i3 are present in the left dentary, with evidence for a developing germ in i2 (Fig 8C). In the right dentary all three incisors are preserved (Fig 8D). Only i1 and i3 show evidence of replacement, in the form of associated replacement germs, of which that of i1 is larger.

**Canines.**   An unusual structure is associated with the root apex of the left maxillary canine (Fig 8A). This structure is located too far dorsally to represent a replacement tooth and is instead interpreted as the last unresorbed remnants of the previous canine, based on its resemblance to a closed root apex. A developing replacement canine is present medial to the functional canine in the right maxilla (Fig 8B). No replacement of the lower canines is evident (Fig 8C and 8D). The root apices of the four functional canines are fully developed.

**Postcanines.**   Nine of a total of 11 postcanines are preserved in the left maxilla, with empty alveoli present at PC5 and PC6 (Fig 8A). A single replacement tooth is preserved in association with the locus of PC10. Only four postcanines of the right maxilla (PC1, PC4, PC5 and PC10) are preserved *in situ* (Fig 8B). An additional postcanine tooth is preserved distal to PC10. This element is interpreted as being *ex situ*, due to the horizontal orientation of the tooth, but may represent the exfoliated PC11/PC12. There are no preserved roots from shed teeth, and there is evidence for three replacement teeth. The replacement associated with the locus of PC7 is the largest, whereas those of PC4 and PC5 are very small.

Twelve erupted postcanines are preserved in the left dentary (Fig 8C). Two replacement teeth are present in the left dentary at the pc9 and pc10 positions, the latter being larger. Eleven functional postcanines and two empty alveoli, at positions pc3 and pc6, are preserved on the right dentary (Fig 8D). Both pc3 and pc12 have replacement teeth below the gingival margin, whereas old roots from previously shed teeth are present at pc3 and pc6. Another root of a shed tooth is preserved labially to pc8. The alveolus of pc8 also preserves the developing crown of a replacement tooth.

### BP/1/4602 (BSL 88 mm)

**Incisors.**   Four incisors are preserved in the left premaxilla (Figs 5A and 9A). Open roots are visible in I2 and I4. The only replacement tooth is associated with I3, whereas the root of the functional I3 shows signs of resorption associated with the replacing tooth. In the right pre-maxilla I1, I3 and I4 are preserved. The roots of the I1 and I4 are open, whereas the root of I3 is closed (Figs 5B and 9A). An *ex situ* crown, possibly shed from I2, is preserved lingual to I3. There is no evidence of replacement activity.

Three incisors are present in each dentary, and in both instances, only i2 shows replacement activity (Fig 5B). On the left, the root of i2 has been completely resorbed (Fig 9C). The crown of the functional tooth is situated higher than those of the neighboring incisors, which indicates that the tooth was about to be shed. The right i2 has not undergone as much resorption as the left i2, such that the crown and root are still connected (Fig 9D). The roots of i2 and i3 appear longer and more slender than that of i1, possibly due to the incomplete development of i1, as the root is still open.

**Canines.**   Both maxillae present a large unerupted replacement canine, in similar developmental stages, anteromedial to the functional tooth (Fig 9A and 9B). A remnant canine root is retained in the right maxilla, mesial to the functional canine. The roots of the functional maxillary canines are closed. Two tooth generations of the left mandibular canine are present (Fig 9C): a functional crown nearing exfoliation with corresponding root remnants, and a moderately developed crown of a replacing canine. The replacement canine has pierced the root of the crown to be shed. In the right dentary, the replacement crown is slightly more developed than the left, and much of the crown of the old functional canine has been resorbed (Fig 9D). The crown to be shed has been displaced such that it no longer lies in the same plane as the remnant root mesial to the erupting tooth. The apex of the replacing crown does not pierce the old crown and is instead directed labially.

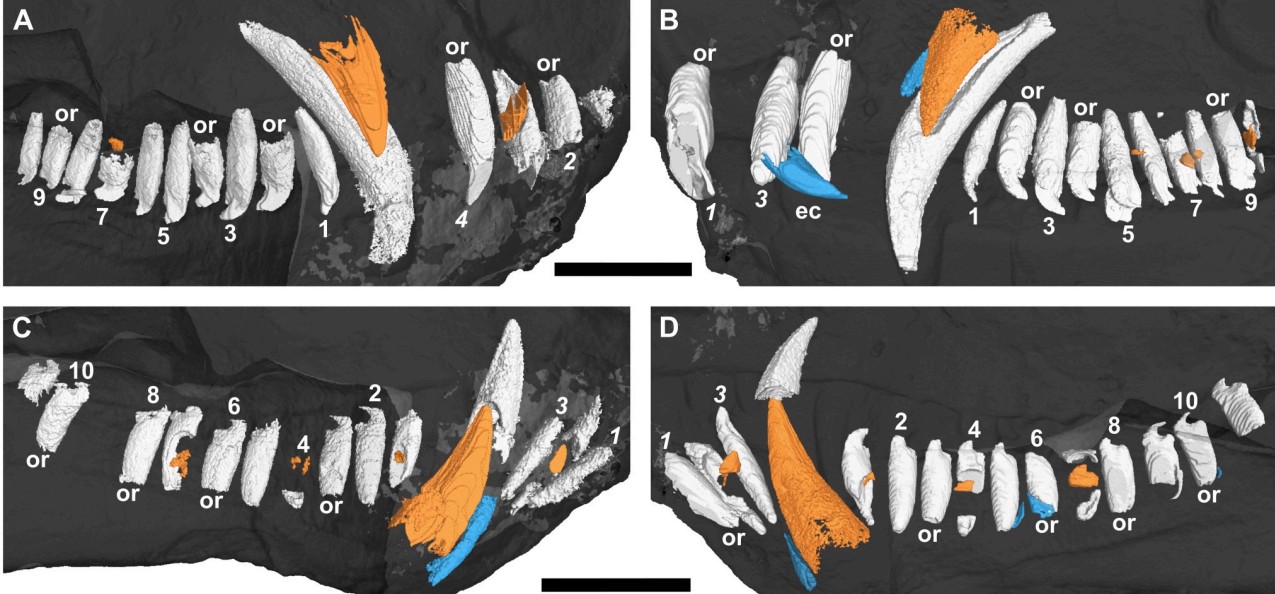

**Fig 9. Three-dimensional rendering of the tooth rows of a subadult *Galesaurus planiceps* (BP/1/4602) in medial view.** (A) Upper left. (B) Upper right. (C) Lower left. (D) Lower right. Replacement teeth in orange, old remnant roots in blue. Abbreviations: ec, exfoliated crown; or, open root. Arabic numerals indicate incisor (italicized) and postcanine positions. Scale bars equal 20 mm.

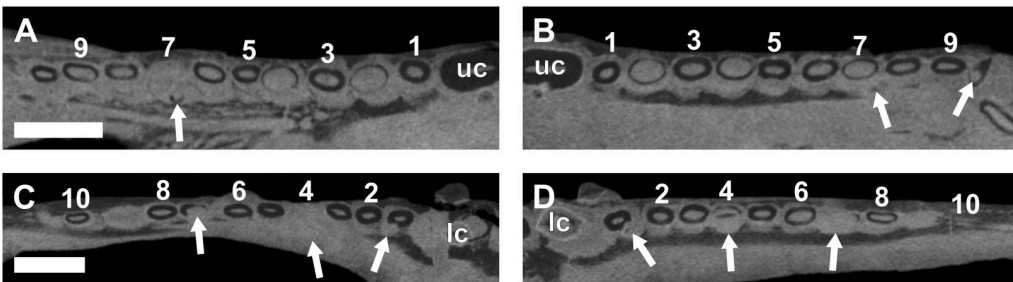

**Fig 10. Virtual horizontal sections through the maxillae and dentaries of a subadult *Galesaurus planiceps* (BP/1/4602).** (A) Left maxillary postcanine series. (B) Right maxillary postcanine series. (C) Left mandibular postcanine series. (D) Right mandibular postcanine series. Abbreviations: lc, mandibular canine; uc, maxillary canines. Arrows indicate crypts of replacement postcanines. Arabic numerals indicate postcanine positions. Scale bars equal 5 mm.

**Postcanines.** In the left maxillary series (Figs 9A and 10A), 10 postcanine teeth are preserved. The root of PC1 is very close to, but does not contact, the root of the functional canine. The roots are open in PC2, PC4 and PC9. A replacement tooth is present lingual to PC7. The root condition of PC7 is similar to that of PC2 and PC4, but is likely caused by resorption of the root by the associated replacement tooth, rather than the incomplete development of the root. The roots of PC3, PC5, PC6, PC8 and PC10 are not yet fully closed, but are more developed than those mentioned above. In the right maxillary series (Figs 9B and 10B), 10 functional postcanine teeth are preserved. The root of PC1 does not contact the functional canine. A small, partially developed replacement crown is preserved between PC5 and PC6. Larger, more developed crowns of replacement teeth are present lingual to PC7 and PC10. Open roots are preserved for PC2, PC4 and PC9. The root of PC10 has been eroded away to the extent that the root and crown are no longer in contact.

There are 11 alveoli in the left mandibular series (Figs 9C and 10C), and nine functional teeth are preserved. An alveolus containing a retained root is situated in the locus of pc4, whereas the alveolus of pc9 is empty. Replacement germs are associated with pc1, pc4 and pc7, the latter being more developed. In the right mandibular series (Figs 9D and 10D), 11 teeth are preserved. The root of pc1 is closed and shows signs of resorption where the root is in close proximity to the replacing canine. There are replacement teeth associated with pc1, pc4 and pc7, the same condition observed on the left. The root of pc9 shows signs of resorption, but no replacement germ is identifiable. There are old root structures preserved distally to pc5 and pc10, and mesially to pc6. Open roots are present in the pc3, pc6, pc8 and pc10 positions in both dentaries.

## NMQR 135 (BSL 94 mm)

There is no mandible preserved for this specimen.

**Incisors.** The crowns of the functional incisors in both premaxillae have been lost. The second alveolus of the left premaxilla is the smallest. In contrast, in the right premaxilla the first has the smallest diameter (Fig 4A). A remnant root is preserved in the first alveolus of the left premaxilla (Fig 4B). There is a small structure lingually between the I3 and I4 of the right premaxilla, which is not paired, and appears to have a mesial ridge. It is interpreted as the crypt of a developing replacement tooth (Fig 4B).

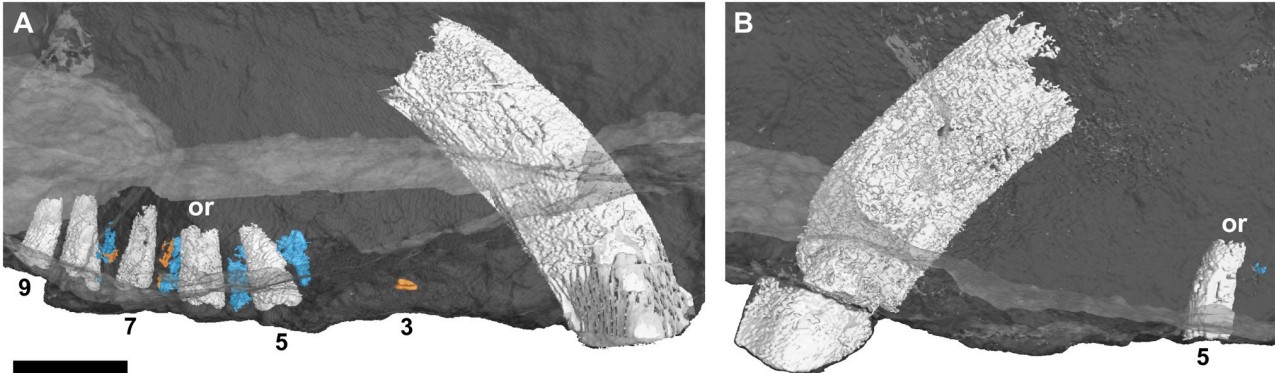

**Fig 11. Three-dimensional rendering of the tooth rows of an adult *Galesaurus planiceps* (NMQR 135) in medial view.** (A) Upper left. (B) Upper right. Replacement teeth in orange, old remnant roots in blue. Abbreviation: or, open root. Arabic numerals indicate postcanine positions. Scale bar equals 5 mm.

**Canines.**   The crowns of the functional canines have been broken. There is no replacement activity associated with either maxillary canine (Fig 11A and 11B). The roots of both canines are open.

**Postcanines.**   Neither maxilla contains a complete postcanine series as several of the teeth have fallen from their alveoli (Figs 4A, 11A and 11B). In the left maxilla, PC1–PC4 are represented by empty alveoli (Figs 4C and 11A), and broken roots are preserved in positions PC5–PC9. A small replacement germ is preserved lingual to the alveolus of PC3. Old root structures are preserved in an alternating pattern with the functional teeth from position PC4 to PC8 (Figs 4C and 11A). These remnant roots show signs of partial resorption and are smaller in root diameter than the associated functional postcanines. No remnant root is preserved between PC8 and PC9, suggesting that the tooth in the PC9 locus was the most recent tooth in the series to erupt, and no replacement had yet taken place at the PC9 (and possibly PC8) locus. Only the root of PC5 is preserved in the right maxillary postcanine series (Figs 4D and 11B). There does not appear to be as many remnant roots preserved between the alveoli of functional postcanines as in the left maxilla.

## NMQR 3542 (BSL ~102 mm)

**Incisors.**   Four incisors are preserved in each premaxilla. Replacement teeth are situated lingual to the left I3 and I4, with the latter being larger (Fig 12A). No replacement activity is recorded in the right premaxilla. Three incisors are preserved in each hemimandible. No replacement teeth are evident (Fig 12B). Remnants of the root of a replaced tooth are associated with the left i2.

**Canines.**   No replacement canines are present in the upper or lower jaws. The maxillary and mandibular canines are open-rooted.

**Postcanines.**   Eleven functional postcanines are preserved in the left maxilla. Replacements are associated with PC3, PC7 and PC8. The right postcanine series also comprises 11 erupted teeth. Of these, PC3, PC4 and PC10 have associated replacement teeth developing. Fourteen functional postcanines are present in the left dentary, with an empty alveolus at pc8. Twelve functional teeth are present in the right lower postcanine series. The only replacement tooth is associated with pc4.

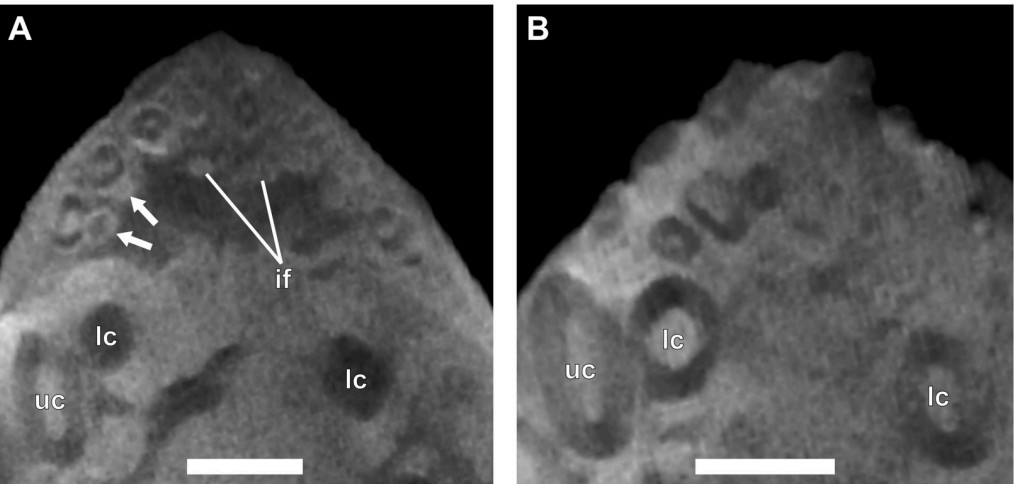

**Fig 12. Virtual horizontal sections through the anterior dentition of an adult *Galesaurus planiceps* (NMQR 3542).** (A) Premaxillary incisors and maxillary canines. (B) Mandibular incisors and canines. Abbreviations: if, incisive foramen; lc, mandibular canine; uc, maxillary canine. Arrows indicate crypts of replacement incisors. Scale bars equal 5 mm.

## BP/1/5064 (BSL 103 mm)

**Incisors.** The broken roots of three incisors are preserved in each premaxilla (Fig 13A and 13B), with the I1 of both sides being absent. Although the crowns are damaged, the largest tooth in each series appears to be I4, based on observable root length and diameter. There is

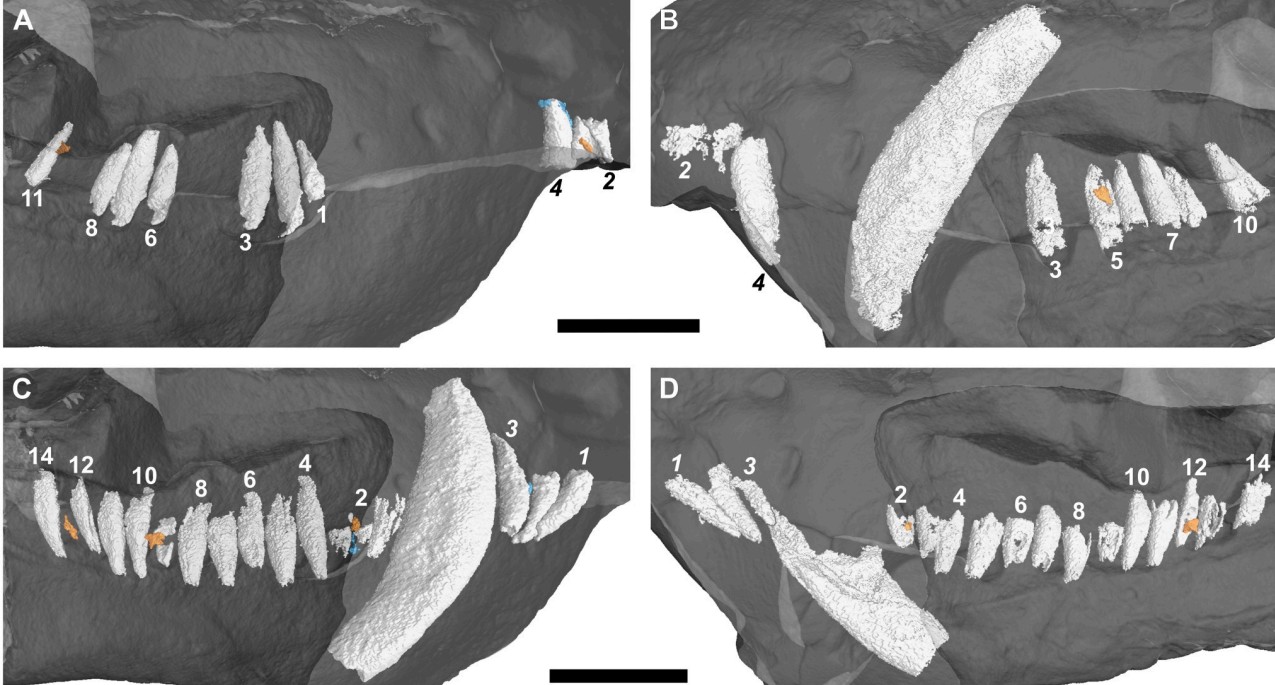

**Fig 13. Three-dimensional rendering of the tooth rows of an adult *Galesaurus planiceps* (BP/1/5064) in medial view.** (A) Upper left. (B) Upper right. (C) Lower left. (D) Lower right. Replacement teeth in orange, old remnant roots in blue. Arabic numerals indicate incisor (italicized) and postcanine positions. Scale bar equals 20 mm.

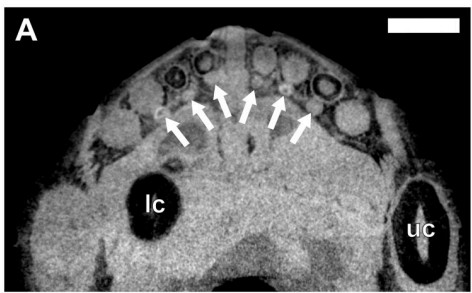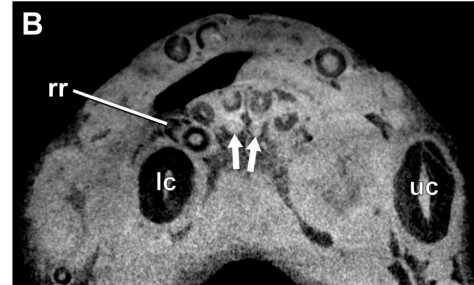

**Fig 14. Virtual horizontal sections through the anterior dentition of an adult *Galesaurus planiceps* (BP/1/5064).** (A) Premaxillary incisors and maxillary canines. (B) Mandibular incisors and canines. Abbreviations: lc, mandibular canine; rr, remnant root; uc, maxillary canine. Arrows indicate crypts of replacement incisors. Scale bar equals 10 mm.

evidence of replacement for the first three incisors (I1–I3) of each premaxilla (Fig 14A). In the left premaxilla, the replacement of I2 is the most developed, and that of I1 the least developed. In the right premaxilla, the replacement I3 is the most developed, whereas the replacements of I1 and I2 are almost equal in size. Crypts associated with the development of replacing teeth are present lingual to the mandibular i1 on both sides, and the left i3 has preserved remnants of an old root (Fig 14B).

**Canines.** The left maxillary canine socket is empty (Fig 13A). The right maxilla (Fig 13B) and left mandible (Fig 13C) preserve almost complete canines, whereas only the root and partial crown of the right mandibular canine is preserved (Fig 13D). The apices of the preserved canine roots are not pointed, suggesting that they were not yet fully developed, and are thus interpreted as being open. There is no replacement activity for either the upper or the lower canines, and there are no old root remnants preserved.

**Postcanines.** In the left maxillary series (Fig 15A) the positions of 11 postcanines can be identified from the preserved teeth and empty alveoli. Developing crypts are evident lingual to the functional PC2, PC3, PC6, PC8 and PC9. The crypt associated with PC8 is the largest, whereas the crypts for PC3 and PC6 are of comparable developmental stages (Fig 15A). The replacement for PC2 is minute in comparison to the others. A small replacement germ is present at the locus for PC11 (Fig 13A). In the right maxillary series (Fig 15B), 11 postcanine loci are present, considering preserved teeth and empty alveoli. Replacement crypts are present lingually to PC1, PC3, PC6 and PC8. A mineralized replacement crown is present lingual to PC5

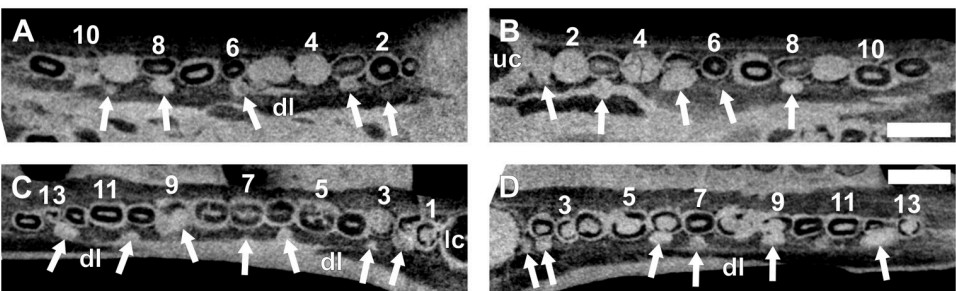

**Fig 15. Virtual horizontal sections through the maxillae and dentaries of an adult *Galesaurus planiceps* (BP/1/5064).** (A) Left maxillary postcanine series. (B) Right maxillary postcanine series. (C) Left mandibular postcanine series. (D) Right mandibular postcanine series. Abbreviations: dl, dental lamina groove; lc, left mandibular canine; uc, right maxillary canine. Arrows indicate crypts of replacement postcanines. Arabic numerals indicate postcanine positions. Scale bars equal 2 mm.

(Figs 13B and 14B). PC1 is poorly preserved and appears to be in contact with the alveolus of the functional canine. The replacement crypt associated with PC3 is the largest, followed by those of PC6 and PC8 (Fig 15B).

In the left mandibular series (Figs 13C and 15C), 14 loci are preserved with an empty alveolus at pc13. The roots of pc9 are partially resorbed, and replacement teeth are situated lingual to pc2, pc9, and in the alveolus of pc13. Replacement crypts are recorded for pc2, pc3, pc6, pc7, pc9, pc10 and pc12. In the right mandibular series (Figs 13D and 15D), 13 alveoli, and 12 teeth are preserved. Replacement crypts are associated with pc2, pc6, pc7, pc9, pc11, pc12 and pc14. A small crypt is situated mesial to what is interpreted as the locus of pc2, however, there is no space for development of this element as the canine has occupied this region of the postcanine series (Fig 15D). The replacement crypt of pc6 is the largest, followed by those of pc2, pc7 and pc9.

## SAM-PK-K10468 (BSL 105 mm)

**Incisors.**   Four functional incisors are preserved in the left premaxilla (Fig 16A), with a replacement tooth preserved lingual to I3. A similar condition is present in the right premaxilla, with an additional replacement tooth present lingual to the I1. Three incisors are preserved in each dentary (Fig 16B). In the left, a replacement tooth is associated with each functional tooth. The replacement lingual to i1 is the smallest, and that lingual to i2 is the largest. Replacement germs are present lingual to the i1 and i3 in the right dentary. The lower incisors of the right are at a more advanced stage of development than those of the left.

**Canines.**   Canines are present in all four quadrants of the buccal cavity (Fig 16), and their roots are open. Remnant canine roots are preserved distal to the functional canines. These teeth are much smaller than the functional canines (comparable in size to the postcanines) and have a closed root morphology.

**Postcanines.**   Nine functional postcanines are present in each maxilla (Fig 17A and 17B). The right maxilla preserves a crypt on the distal margin of the postcanine series, indicating that a tooth was beginning to develop in the PC10 position (Fig 17B). There is good synchrony of the replacement patterns of the two maxillae, with replacement activity present at positions PC2, PC4 and PC5 in both maxillae, with an additional replacement at PC8 in the right maxilla. In both series, the root of the first tooth appears to have been partially resorbed by the developing canine.

The left lower postcanine series comprises nine erupted teeth and two empty alveoli, whereas the right series includes 10 erupted teeth (Fig 17C and 17D). Both hemimandibles

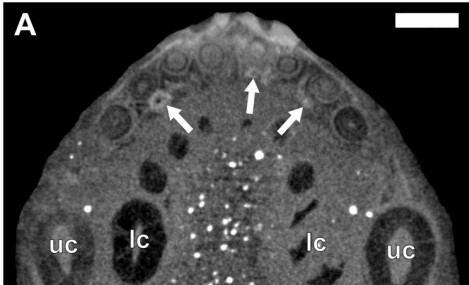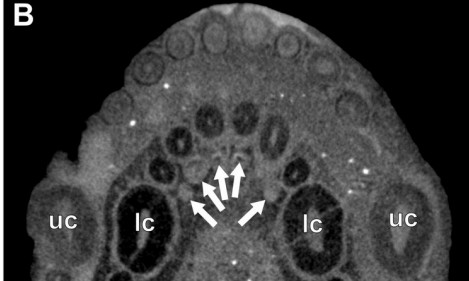

**Fig 16. Virtual horizontal sections through the anterior dentition of an adult *Galesaurus planiceps* (SAM-PK-K10468).** (A) Premaxillary incisors and maxillary canines. (B) Premaxillary incisors and maxillary canine (lateral), and mandibular incisors and canines (medial). Abbreviations: lc, mandibular canine; uc, maxillary canine. Arrows indicate crypts of replacement incisors. Scale bar equals 10 mm.

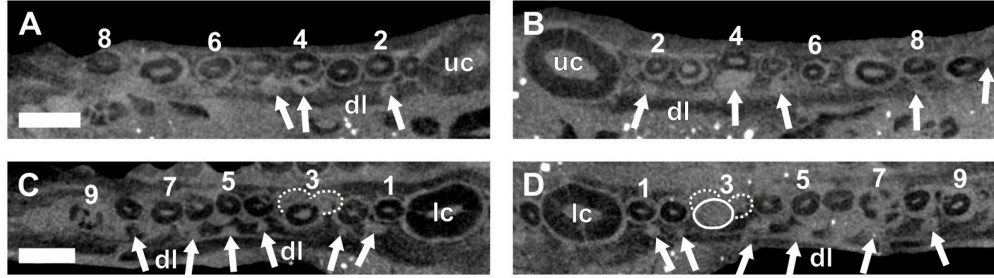

**Fig 17. Virtual horizontal sections through the maxillae and dentaries of an adult *Galesaurus planiceps* (SAM-PK-K10468).** (A) Left maxillary postcanine series. (B) Right maxillary postcanine series. (C) Left mandibular postcanine series. (D) Right mandibular postcanine series. Note the replacement tooth at the third locus (C and D) appears to be occupying the loci of two exfoliated teeth. Abbreviations: dl, dental lamina groove; lc, mandibular canine; uc, maxillary canine. Arrows indicate crypts of replacement postcanines. Solid line indicates the outline of replacement postcanine (pc3); dashed lines indicate empty alveoli associated with the pc3 locus. Arabic numerals indicate postcanine positions. Scale bars equal 4 mm.

have crypts associated with the pc1 and pc2 positions. Of particular interest is the region associated with the erupting pc3 in the left dentary. This tooth is larger than the surrounding teeth and appears to be filling the position of two previous loci (Fig 17C). A similar condition is observed in the right dentary, but the alveolus is empty (Fig 17D). Additional replacement teeth are recorded in the pc4, pc5, pc6 and pc8 positions of the left dentary, and the pc4, pc5, pc7 and pc8 positions of the right. A good degree of synchrony is observed between the anterior postcanine loci of the two hemimandibles, but developmental stages in positions distal to pc5 become asynchronous.

## NMQR 860 (BSL 114 mm)

**Incisors.** Each premaxilla preserves four incisors (Fig 18A). On the left, I2 is represented by a large empty alveolus. The crown of the right I4 is damaged, and the position of the root in the alveolus suggests that the tooth was in the process of being lost. Whether this exfoliation was due to a natural or taphonomic process is unclear. A resorption pit associated with the left I4 contains a mineralized germ of the replacement tooth. There are three lower incisors preserved in each mandible, and no replacement teeth are visible. The roots of the i3 in each series

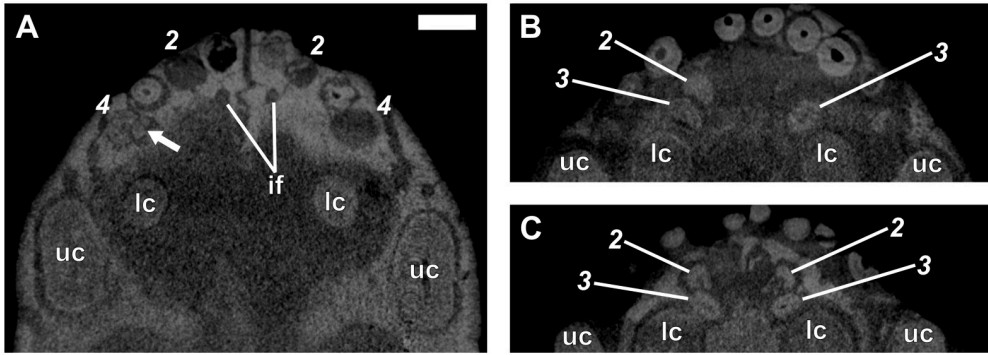

**Fig 18. Virtual horizontal sections through the anterior dentition of an adult *Galesaurus planiceps* (NMQR 860).** (A) Premaxillary incisors and maxillary and mandibular canines. (B) Premaxillary incisors and maxillary canines (lateral) and mandibular incisors and canines (medial). (C) Mandibular incisor roots. Abbreviations: if, incisive foramen; lc, mandibular canine; uc, maxillary canine. Arrow indicates crypt of replacement incisor. Arabic numerals (italicized) indicate incisor positions. Scale bar equals 4 mm.

**Fig 19. Virtual horizontal sections through the maxillae and dentaries of an adult *Galesaurus planiceps* (NMQR 860).** (A) Left maxillary postcanine series. (B) Right maxillary postcanine series. (C) Left mandibular postcanine series. (D) Right mandibular postcanine series. Abbreviations: lc, mandibular canine; uc, maxillary canine. Arrows indicate crypts of replacement postcanines. Arabic numerals indicate postcanine positions. Scale bars equal 5 mm.

lie lingual, almost posterior, to those of the neighboring i2. In contrast, the crowns of all three elements lie in the same plane in the dental arcade.

**Canines.** Functional canines are preserved in all four quadrants of the buccal cavity. No replacement canines are preserved, and the root morphology of all four canines is open.

**Postcanines.** Ten postcanines are preserved in the left maxilla (Fig 19A), with small replacement teeth located lingual to PC3 and PC8. The right maxilla contains a postcanine series comprising 10 teeth (Fig 19B). Replacements are associated with PC2 and PC5. Small crypts are located lingual to PC6 and PC8, but neither contains mineralized tooth tissue. There are 13 mandibular postcanines in the left dentary (Fig 19C). Several alveoli are empty, including pc3, pc6 and pc9. Replacement germs are present lingual to the alveoli of pc6 and pc9, and what appear to be the remains of the tooth root is preserved in pc3. At least 12 postcanine loci are evident in the right dentary (Fig 19D). The first 10 positions show erupted teeth, whereas those of pc11 and pc12 are empty alveoli. A retained root is situated between the first and second erupted teeth, suggesting that the full series may have been 13 teeth, as is the condition in the left dentary.

## Anatomical specimens

Several specimens of *Galesaurus planiceps* were studied for gross morphological patterns. Because most of these specimens have not been fully prepared, or are not very well preserved, little information regarding the tooth replacement patterns could be added from direct observation of the specimens. One noticeable exception is SAM-PK-K1119, a specimen that was subjected to acid preparation prior to this study [20, 71].

## SAM-PK-K1119 (BSL 72 mm)

**Incisors.** Four teeth are preserved in each premaxilla. A pit is present lingual to the right I3, which may be indicative of an associated replacement tooth (Fig 20A). Three incisors are preserved in each dentary, with resorption pits lingual to the left i3 and between the right i2 and i3 (Fig 20B).

**Canines.** An erupting replacement canine is present posterolingually to the functional canine in both dentaries (Fig 20B). On the right, the tips of both the functional and replacing crowns are broken. In contrast, the crowns are well preserved in the left dentary. The tip of the replacing canine is not in contact with the functional tooth. No replacement activity can be seen for the upper canines.

**Postcanines.** Seven postcanine teeth are present in the left maxilla (Fig 20A). There is a space between the preserved second and third tooth, which appears to be an empty alveolus

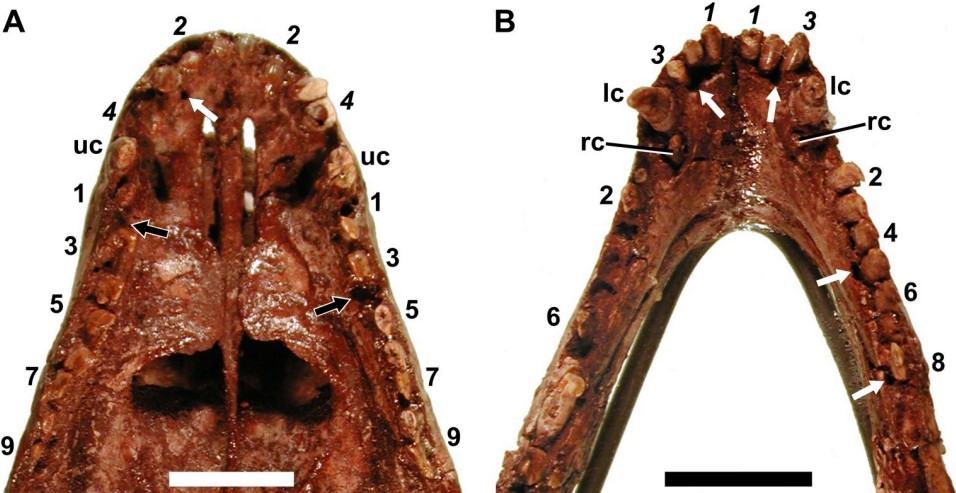

**Fig 20. Dentition of a subadult *Galesaurus planiceps* (SAM-PK-K1119).** A, upper dentition; B, lower dentition. Abbreviations: lc, mandibular canine; rc, replacement lower canine; uc, maxillary canine. White arrows indicate resorption pits; black arrows indicate spaces between erupted teeth. Arabic numerals indicate incisor (italicized) and postcanine positions. Scale bars equal 10 mm.

corresponding to PC4. The erupted postcanine count on the right maxilla is seven, however, there is a gap between the first and second tooth, which is of sufficient size to have once held a tooth (Fig 20A).

The postcanine series is poorly preserved in the left dentary (Fig 20B). The crown of pc1 is broken at the level of the bone, and the alveoli of pc3–pc5 and pc7 are empty. Evidence for at least two, possibly three, additional teeth distal to the alveolus of pc7 is visible. These crowns are damaged, with the potential of pc9 being displaced. Eight postcanine positions are preserved in the right dentary. The crown of pc1 is broken, and pc7 is represented by an alveolus with an unerupted replacement tooth. The alveoli surrounding pc5 and pc8 are wider than those of the other teeth. In addition, the pc2, pc5 and pc8 stand out of their respective alveoli such that the tooth neck is visible. It is suggested that these teeth were in the process of being exfoliated. In both dentaries, pc1 has a noticeably smaller root diameter than the other teeth of the series.

## Discussion

### Tooth morphology and number

**Anterior dentition.**   The dental formula of four upper and three lower incisors, as is present in most basal Permian cynodonts, also applies to *Galesaurus planiceps* [54, 72]. This dental formula is retained in the majority of the Early to Middle Triassic cynodonts [15, 73]. The crowns of the upper and lower incisors are conical with the upper incisors being slightly more recurved than the lowers.

The upper incisors of all specimens are evenly spaced, and the roots are round in cross-section. In contrast, the crowns of the lower incisors are evenly spaced, but the roots are tightly packed. The lower incisors of the larger specimens (BP/1/5064 and NMQR 860) show more pronounced crowding of the roots of the i2 and i3 than in the smaller specimens. In both specimens, the root of the left i3 is positioned lingual to that of i2, whereas the erupted crowns are situated laterally to one another (Fig 14B). In NMQR 860, the roots of i2 and i3 are in such

close association that it appears that partial resorption of the root of i2 has taken place (Fig 18B and 18C). The crown of the erupted left i3 in BP/1/5064 is positioned more lingual than that of the neighboring i2 (Fig 14B). This is interpreted as the result of i3 having not migrated completely into its natural position. The presence of a retained root labial to the erupting i3 is further evidence that the process of replacement at the locus was not yet complete (Fig 14B). This crowding of the incisors led van Hoepen [51] to originally interpret the number of lower incisors in TM 24 (BSL ~71 mm) as two.

A single functional canine tooth is present in each quadrant of the buccal cavity. The lateral ridges of the canines resemble the 'cutting edges' present in the 'incisiform dentition' of the Nile crocodile (*Crocodylus niloticus*) [74, 75]. The maxillary canines of *Galesaurus* do not bear the dorsoventrally directed facets on the distal surface of the canine, which are present in *Thrinaxodon* [44]. These structures in *Thrinaxodon* are not interpreted as occlusal wear surfaces, as the mandibular canine passes mesial to the maxillary canine, suggesting that they may rather represent an adaptation related to feeding. Furrowed canine teeth, on the other hand, are present in *Cynosaurus* and *Progalesaurus* [7] and may be an adaptation to strengthen the tooth crown.

The canine roots of specimens RC 845, BP/1/4714 and BP/1/4602 are closed, whereas those of the large specimens NMQR 135, NMQR 3542, BP/1/5064, SAM-PK-K10468 and NMQR 860 are open. Interestingly, the remnant roots of the shed maxillary canines of SAM-PK-K10468 have a closed morphology. An additional specimen, AMNH FARB 2227 (BSL 79 mm) has maxillary canines with root apices that are nearly closed [50], and approximate the developmental stages of the similarly sized BP/1/4714 (Fig 8A and 8B) and BP/1/4602 (Fig 9A and 9B). This indicates that skulls with a BSL ≤ 88 mm have closed canine roots, whereas the larger skulls (BSL ≥ 94 mm) have open roots, a condition previously reported in various dicynodont taxa [41, 76–83]. This change in canine root morphology is accompanied by the cessation of replacement of the canines and coincides with the transition from subadult to adult skull size [13].

**Postcanines.** The number of postcanines recorded in the study sample varies from nine to 12 maxillary, and 11 to 15 mandibular teeth, with subadult and adult specimens having the largest number of teeth. The number of postcanine teeth agrees with those reported in previous descriptions [1, 51–55] and confirms the idea that the length of the postcanine series varies with basal skull length (e.g., Figs 21 and 22). Specimen FMNH PR 1774 (WM 1563 in Rigney [63]), which has seven teeth in each maxilla, has the least number of upper postcanines. The largest number of upper postcanines is in TM 83 with 12 teeth, although BP/1/5064 shows evidence for the development of a twelfth postcanine in both maxillae.

RC 845, the smallest specimen included in the study, has the fewest mandibular postcanines, with only 11 in each dentary. The larger BP/1/5064 exhibits the most postcanines, with evidence for 14 functional postcanines in the left, and 13 in the right dentary. The functional teeth of the right dentary are, however, interpreted as representing loci pc2–pc14, as a crypt is evident mesial to the position of pc2 (Fig 15B).

In the smaller specimens that were μCT-scanned, the first tooth of the postcanine series has a simplified crown morphology, with a straighter mesial cusp and an incipient distal cusp (Fig 1B). This morphology of PC1 was first described and illustrated by Broom [53, 54]. The tooth is also mesially inclined and is smaller than the following teeth in the series. This gives an overall more conical, caniniform appearance compared to PC2. A comparable condition has been described for the fourth upper incisor of *Thrinaxodon* [44]. This simplification of the fourth incisor to a canine-like morphology in *Thrinaxodon* has been suggested to be related to the manner in which the animal captured prey [44].

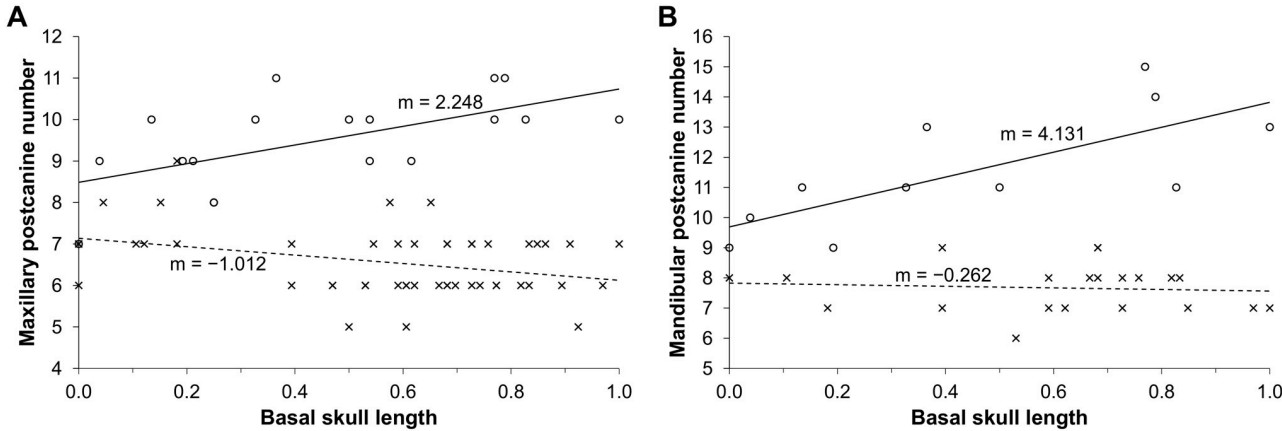

**Fig 21. Number of postcanines in *Galesaurus planiceps* (○) and *Thrinaxodon liorhinus* (×).** (A) Maxillary dentition. (B) Mandibular dentition. X-axis represents the normalized basal skull length (BSL) and Y-axis the number of postcanines present. Solid trend line, *Galesaurus planiceps*; dashed trend line, *Thrinaxodon liorhinus*.

All teeth distal to PC2 exhibit the bicuspid crown morphology typical of *Galesaurus*. Additionally, small mesial accessory cusps have been described in the more distal postcanines (PC5–PC8) of three small specimens of *Galesaurus* [13] (Fig 1C), and the presence of these accessory cusps is attributed to ontogenetic variation. This supports the idea that the morphological variation of the postcanine crowns seen in *Galesaurus* may be due to ontogenetic variation, rather than taxonomic variation.

In *Thrinaxodon* there is a reduction in the number of postcanine teeth in the series with each successive replacement [41, 84, 85]. *Galesaurus*, on the other hand, increases the length of the postcanine series with size (Fig 21). This condition has a superficial resemblance to that of *Diademodon*, but the manner in which it is achieved in *Galesaurus* is not the same. In *Diademodon*, the mesial-most tooth of the postcanine series had no replacement [45, 46, 48, 49]. Once the first postcanine was exfoliated, the alveolus became filled with bone, which was subsequently remodeled to the extent that the filled alveoli mesial to the postcanine series are indistinguishable from the surrounding bone [49]. Each successive exfoliation of the first postcanine results in the distal shift of the postcanine series by one locus, and an increase in size of

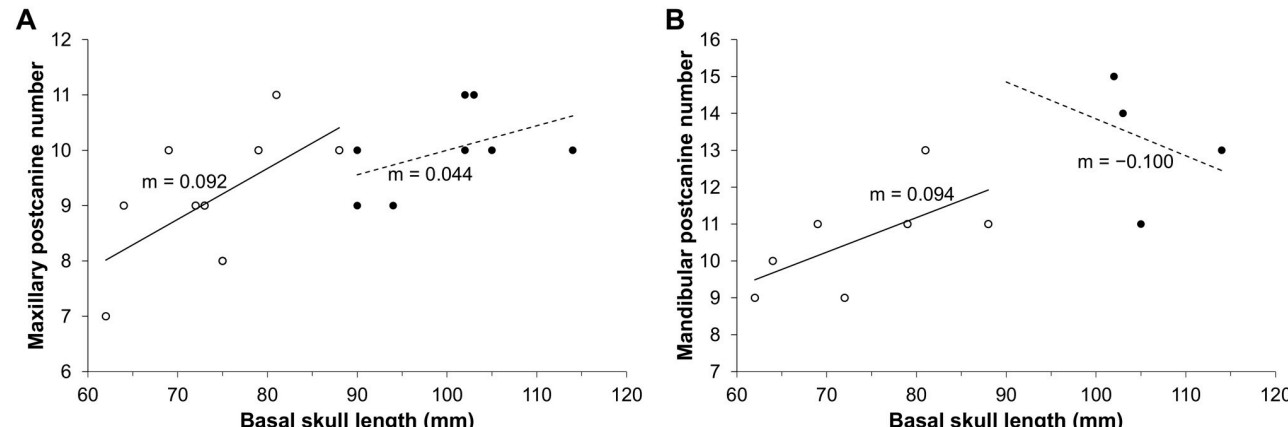

**Fig 22. Number of postcanines in subadult (○) and adult (●) *Galesaurus planiceps*.** (A) Maxillary dentition. (B) Mandibular dentition. X-axis represents the basal skull length (BSL) and Y-axis the number of postcanines present. Solid trend line, subadults; dashed trend line, adults.

the diastema between the canine and postcanine series. This is not observed in *Galesaurus*, as replacement crypts are present lingual to the first postcanine in the maxilla of BP/1/5064 (Fig 15B), and the dentaries of BP/1/5064 (Fig 15D) and SAM-PK-K10468 (Fig 17C and 17D).

Estes [85] wrote that the tooth morphology in *Thrinaxodon* was similar in specimens of various sizes, but juvenile specimens of *Thrinaxodon* had more postcanine teeth (seven) when compared to adults (six postcanines). Abdala et al. [44] considered the typical number of postcanines in adult *Thrinaxodon* to be six maxillary and seven–eight mandibular teeth. It was demonstrated that crown morphologies of the upper postcanines were simpler than those of the lower teeth, with the most complex maxillary postcanines being tricuspid, lacking a cingular collar. Abdala et al. [44] also demonstrated a reduction in the mandibular postcanine crown complexity with increased size. Specimens of *Thrinaxodon* with a BSL ≤ 80 mm have multicusped crowns, whereas larger specimens only have tricuspid crowns. This condition is similar to that observed in *Galesaurus*.

The postcanine crown morphology of the contemporaneous *Progalesaurus* (SAM-PK-K9954) is more complex than that of *Galesaurus*, bearing well developed mesial accessory cusps, and several distal accessory cusps [7]. Additional mesial accessory cusps have recently been observed in two subadult specimens of *Galesaurus* [13]: BP/1/4597 (BSL ~70 mm) and NMQR 3716 (BSL 75 mm). Here we report a third subadult specimen (SAM-PK-K9956, BSL 73 mm), which has mesial cusps on PC7 and PC8 (Fig 1C). No such accessory cusps were observed in adult specimens, suggesting a simplification of crown cusp morphology with each successive replacement, as has been reported for *Diademodon* [48, 49] and *Nanictosaurus* [86]. The specimen of *Progalesaurus* is much larger (BSL 93.5 mm) than any of the *Galesaurus* specimens reported with mesial accessory cusps on the postcanines. At present *Progalesaurus* is represented by one specimen, and consistently falls as the sister taxon to *Galesaurus* in phylogenetic analyses [7, 8, 10, 73, 87–89].

## Tooth replacement

The pattern of tooth replacement observed in *Thrinaxodon* [40, 44] corresponds with Edmund's [90, 91] Zahnreihe theory, whereby teeth are replaced in successive waves that move through the jaw in a back to front direction. In *Galesaurus*, replacement of the anterior teeth (incisors and canines) matches that of *Thrinaxodon*, whereas the pattern of the postcanine replacement differs.

In *Galesaurus*, replacement teeth are always positioned lingual to the functional tooth they are to replace. As the new tooth develops, it moves labially, etching at, and causing the resorption of the root of the functional tooth. Once the root has been sufficiently eroded through this process, the crown is shed [92]. A similar sequence of resorption of the functional tooth root coinciding with the development of the replacing crown in the postcanine series of *Galesaurus* is described in the Diadectomorpha [93].

**Incisors.** A typical pattern of alternating replacement is recorded for both the upper and lower incisor compliments in the sample. A good example of this for the upper series is seen in the right premaxilla of RC 845 (Fig 2B), where the even-numbered incisors (I2 and I4) are at a later developmental stage than the odd-numbered incisors (I1 and I3). The odd-numbered incisors show only the crown up to the approximate level of the tooth neck, whereas the even-numbered incisors have more prominently developed roots. The condition is less obvious in the left premaxilla of RC 845, but again the even-numbered teeth are more developed. A noticeable exception is observed in BP/1/5064, which has replacement crypts associated with the first three incisors (I1–I3) in each premaxilla. In both sides, the third crypt is larger than the others, which are all of similar size.

An alternating replacement pattern is evident in the lower dentition of several specimens. In RC 845 and BP/1/4602, replacement incisors are only associated with the even-numbered teeth (i2), with a fair amount of synchrony of the developmental stages in each hemimandible. Asynchronous replacement is, however, evident in BP/1/4714 (Fig 8C and 8D), where replacement germs are associated with i2 in the left dentary, and i1 and i3 in the right.

It was not possible to determine which direction the replacement wave travelled from the present sample. However, in specimens that show evidence of two or more replacement incisors in a premaxilla (e.g., RC 845, BP/1/5064 and SAM-PK-K10468), the distal teeth of the odd and even-numbered waves tend to be larger than the mesial teeth (i.e., I3 larger than I1, and I4 large than I2). This suggests a back-to-front movement of the replacement wave, but we cannot rule out the idea that the mesial tooth may be smaller, because it represents the initiation of the next replacement wave. If this is correct, then it would support the replacement waves moving in a front-to-back direction.

**Canines.** Specimen RC 845, which has the smallest BSL in the study, is considered a subadult [20, 22]. The advanced stage of development of the maxillary replacement canines (Fig 2A and 2B), and the presence of remnant roots associated with three of the four canines (Fig 2A, 2C and 2D), indicates that the teeth present in RC 845 do not represent the earliest canine dentition of the individual. Specimen BP/1/4602 is the only skull of *Galesaurus* that exhibits replacement of all four canines simultaneously (Fig 9). This degree of canine replacement was present in the whole sample of μCT-scanned *Thrinaxodon* specimens [44], with the exception of the largest specimen, BP/1/5905 (BSL 87 mm), where replacement maxillary canines are evident, but there are no replacement canines in the mandible. This is similar to the condition reported for the subadult *Galesaurus* RC 845 (Fig 2).

No replacement of the canines was documented for specimens of *Galesaurus* with a BSL of 90 mm or more. Replacement of the canines thus appears to be restricted to juvenile and subadult *Galesaurus*, whereas open rooted canine morphology with no signs of replacement activity is observed only in adult specimens larger than 94 mm (e.g., NMQR 135). This suggests that there are a finite number of replacement generations for the canines in *Galesaurus* and that the final 'permanent' generation has a different root morphology to that of 'deciduous' generations. This differs from the condition seen in *Thrinaxodon* where continual replacement of the canine teeth has been recorded well into adulthood [44].

The condition of continuously growing, open-rooted canines is a feature previously considered to have been a unique specialization of dicynodonts amongst the therapsids [79]. It is important to note that the caniniform teeth (tusks) of dicynodonts are not homologous to the canine teeth of the Theriodontia [78, 80, 94]. Among non-mammaliaform cynodonts open canine roots have previously been hypothesized to be present in the Middle Triassic traversodontid *Andescynodon* due to hypertrophy of the canines [95].

Several examples of open-rooted dentitions, or 'tusks,' exist in extant mammalian lineages. Proboscideans [96], dugongs [97], rodents and lagomorphs [98] have open-rooted incisors, whereas walruses [99], beaked whales [100, 101], narwhals [102], and hippopotamuses [103] have open-rooted canines. In many of the examples presented, these teeth are adapted to serve a secondary purpose not necessarily related to feeding, such as intraspecific competition (e.g., elephant and hippopotamus [104]) or even as a sensory organ (e.g., narwhal [105]). Often the teeth are also influenced by sexual dimorphism [98] (e.g., primates and narwhal [105, 106]). Sexual dimorphism has recently been suggested to have occurred in *Galesaurus* [13], however, open-rooted canines are present in both 'female' (e.g., NMQR 135 and NMQR 3542) and 'male' morphotypes (e.g., BP/1/5064, SAM-PK-K10468 and NMQR 860).

It is hypothesized that by the time an individual had attained a basal skull length of ~90 mm, the rate of mineralization of the canine teeth had slowed to such an extent that the root

apex would not have fully developed prior to the animal's death. Possible reasons for such sudden slowing of canine tooth formation and cessation of replacement may be related to the attainment of skeletal maturity. Sexual maturity has been inferred to have been attained at a larger size in *Galesaurus* compared to *Thrinaxodon* [13]. Parental care has also been demonstrated to have occurred in *Galesaurus* [22]. Changes in behavior, such as spending less time hunting/foraging and more time caring for young may have resulted in physiological changes such as a slower rate of mineralization of the canine teeth in larger specimens.

There is no evidence of multiple canine tooth families in *Galesaurus*, such that at any one time only a single functional canine is present in each quadrant of the mouth. In contrast, several specimens of *Thrinaxodon* present evidence for multiple tooth families (e.g., BP/1/5372, TM 80A, BP/1/7199 and TM 180) [44]. In the maxilla of these specimens, both the replacing canine and retained roots from the previous canine are positioned mesial to the functional canine. This arrangement has been interpreted as being caused by the migration of the functional canine [44]. This condition is not seen in *Galesaurus*, as in all the μCT-scanned specimens the roots of the previous maxillary canines have been almost completely resorbed. Similarly, in the mandible of several specimens of *Thrinaxodon* (e.g., BP/1/5372, TM 80A and BP/1/7199) the replacement canine and remnant roots are both situated distal to the functional canine. A similar hypothesis has previously been proposed for the migration of the mandibular canines [42]. Some small *Thrinaxodon* specimens: BP/1/1376 (BSL ~30 mm), TM 1486 (BSL ~33 mm) and SAM-PK-K10016 (BSL 42 mm) have replacement maxillary canines distal to the functional canine, a condition described as being rare in this taxon [44]. In addition to the distal eruption of the replacing canine, TM 1486 is unusual in that it also has a smaller replacement bud lingual to the root of the functional canine [44], suggesting the presence of multiple replacement generations. Eruption of a replacement canine lingual to the functional tooth has been described in the basal cynodont *Procynosuchus* [107, 108], as well as the Gorgonopsia and Therocephalia [109].

The data from the smallest *Galesaurus* specimen scanned (RC 845, BSL 69 mm) indicates that the development and eruption of the mandibular canines took place earlier than that of the maxillary canines (Fig 2). This interpretation also holds true for the condition seen in SAM-PK-K1119 (BSL 72 mm), where the replacement mandibular canines have already erupted, and the maxillary canines are not yet visible (Fig 20).

The largest *Thrinaxodon* specimen scanned by Abdala et al. [44] that shows replacement of the maxillary canines (BP/1/5905) has a BSL of 87 mm and represents an adult approximately 91% of the size of the largest specimen in their study (SAM-PK-K1461, BSL 96 mm). In contrast, the largest specimen of *Galesaurus* to show replacement of the canines (BP/1/4602) is only 77% of the size of the largest *Galesaurus* specimen (NMQR 860) and is considered to be a subadult [13]. Abdala et al. [44] proposed a higher rate of replacement of the canines in juvenile specimens of *Thrinaxodon*, noting two instances of multiple replacement generations occurring at the same time.

**Postcanines.**   It has been demonstrated in both *Thrinaxodon* [44] and *Diademodon* [49] that replacement of the anterior teeth in the postcanine series ceases, and new teeth are added to the distal margin of the postcanine series, causing a distal shift in the series. This distal shift in the postcanine series is also observed in *Galesaurus*, however, the presence of replacement crypts mesial to the first functional postcanine in the right dentary of the adult specimen BP/1/5064 (Fig 15D) suggests that the mesial-most elements of the postcanine series may have undergone at least one replacement.

A brief account of the tooth replacement in a juvenile *Galesaurus* specimen (FMNH PR 1774) was given by Rigney [63]. A replacement germ is present for the eighth postcanine in the left dentary. Owing to the greater length of the third and fifth postcanines, Rigney [63]

deduced that an alternating pattern of replacement was present in the postcanines series. Furthermore, Rigney [63] described two canals in the dentary lingual to the postcanine series. The description of these canals is similar to that of the longitudinal grooves described in the maxilla and dentary of *Thrinaxodon* [40]. Crompton [40] proposed that these grooves housed the dental lamina. Similar lingual grooves extending the length of the postcanine series are evident in two adult specimens of *Galesaurus* (BP/1/5064 and SAM-PK-K10468, Figs 15 and 17).

As an individual increased in size, additional postcanine teeth were added to the dental series distally. In contrast, *Thrinaxodon* tends to show a reduction in the number of postcanines in both the maxillary and mandibular series at the transition from juvenile (BSL ≤ 42 mm) to subadult (BSL ~56–68 mm) [16]. The number of postcanines in *Thrinaxodon* specimens larger than 56 mm still shows some variability (Fig 21), but stabilizes at 6–7 elements in the maxillary, and 7–8 elements in the mandibular series [44].

The apparent addition of postcanine teeth mesial to the series in one of the largest specimens of *Galesaurus* (BP/1/5064, Fig 15B and 15D) lends support to the Zone of Inhibition theory [110, 111] used to describe sequences of tooth replacement. However, a more plausible explanation is that the crown of the functional PC1 had been exfoliated due to the resorption of the root by the maxillary (Fig 15B) and mandibular (Fig 15D) canines. Such a condition was proposed for the postcanine dentition of *Thrinaxodon* [44], and further suggests that the PC1 in RC 845 is not homologous to PC1 in BP/1/5064.

The waves of replacement in the maxillae of *Galesaurus* are not as clear as those in the mandibles. This is similar to the findings for *Thrinaxodon* [44], where an alternating replacement pattern was well documented in the mandibular postcanine series, whereas the replacement pattern of the maxillary series was not easy to identify. However, it appears that there are two replacement waves that pass down the hemimandible simultaneously: one affecting even-numbered teeth, and the second affecting odd-numbered teeth. In addition, it appears that these two waves originate in each hemimandible alternately, such that odd-numbered teeth on one side may be more advanced than the odd-numbered teeth of the other side, whereas the converse is true for the even-numbered teeth. Instead of a replacement tooth being associated with every second tooth in the series, replacements are associated with every third tooth in the mesial region of the series, i.e., pc1, pc4 and pc7. The replacement waves apparently run from back to front as the replacement pc7 is more developed than that of pc4, with pc1 being the least developed. Specimen BP/1/4602 (Fig 9C and 9D) shows the best example of this. However, this is based on the assumption that the mesial and distal teeth belong to the same replacement wave.

In at least one specimen (SAM-PK-K10468), there is indication of a single replacement tooth filling the loci of two erupted teeth (locus 3 in Fig 17C and 17D). This may explain the variation in the number of postcanines in the series seen in some specimens having similar or very close BSLs. This anomaly may be due to the alternating waves of replacement, and could explain the complexity seen in the patterns of the replacement waves. There appears to be a slowing in the rate of replacement of the postcanines as BSL increases, such that the waves replacing odd and even-numbered postcanines may begin to move along the series at differing rates. This may account for the condition of two teeth adjacent to one another both being in the process of replacement (e.g., BP/1/5064 and SAM-PK-K10468).

It was not possible to determine the number of replacement waves present in either postcanine series of *Galesaurus*. However, in the maxilla, it is inferred that a minimum of five replacement waves took place, as there is an increase in the number of postcanines from the smallest (FMNH PR 1774, seven maxillary postcanines) to the largest (TM 83, 12 maxillary postcanines) by a count of five elements. Assuming that each successive wave would add one element to the distal margin of the postcanine series, five replacement waves are estimated.

However, it is likely that the first postcanine tooth was shed to accommodate the expanding alveolus of the canine [44], especially in juveniles. Thus, the increase in the number of postcanine teeth with an increase in skull length suggests that teeth are added to the distal margin of the postcanine series faster than they are exfoliated from the mesial margin of the same series. Given the perceived alternating pattern of three loci, it is plausible to consider that the first locus of the series ceases to be active after two or three tooth generations. Thus, every three replacement waves would comprise a complete 'replacement cycle' of the entire postcanine series, resulting in a net gain of two postcanine teeth (+ 1 distal locus per wave and − 1 mesial locus per replacement cycle). Specimens with a BSL smaller than 81 mm appear to adhere to this proposed replacement triplet of + 3: − 1 (i.e., two teeth are gained after one complete replacement cycle), whereas in adult specimens the pattern is not as clear, and may more closely resemble + 2: − 1 due to the slowing of the replacement waves.

In the sample, there is a decline in the number of maxillary postcanine teeth (from 11 to 9) in specimens with a BSL between 80 and 90 mm (Fig 22A), which represents the transition between subadults and adults. This corresponds to the skull size at which replacement of the canines is no longer recorded. Similar observations were not possible for the mandibular postcanine series, as there were no specimens in the BSL range of 90 to 100 mm that have accessible mandibular teeth (Fig 22B). Interestingly, there is a decline in the number of mandibular postcanine teeth with increased BSL in specimens larger than 102 mm.

For the maxillary series, it is interpreted that at a BSL of approximately 80 to 90 mm the replacement rates of the postcanine series slows such that two replacement waves are present in the tooth row at the same time. This may account for the perceived increase in replacement activity based solely on observations of the number of crypts being present in larger specimens (e.g., BP/1/5064, Fig 15). Whereas the + 3: − 1 replacement triplet may still apply, it is likely masked by the presence of the first wave of the next triplet occurring concurrently with the last wave of the preceding triplet, leading to observations that do not follow the original model proposed. Such instances are observed in the maxilla at the transition from 90 to 94 mm, and 103 to 105 mm. These account for the first tooth of the leading wave having been shed, but as the leading wave was still developing, it has yet to reach the point of adding an additional tooth to the series.

It has been demonstrated in the Gorgonopsia and Therocephalia [41, 109] that replacement of the postcanines ceased first, followed by the canines, and lastly the incisors. However, in *Galesaurus* replacement of the canines ceases at the attainment of adult size (BSL ~90 mm), and replacement of the incisors and postcanines continues even in the largest specimens.

We present a replacement model for the postcanine dentition of *Galesaurus* in Fig 23. It should be mentioned that the model has two limitations: (1) it assumes a constant rate of development of replacing teeth through ontogeny, and (2) it assumes that each replacing tooth erupts directly into the locus of the associated functional tooth. The latter has been observed to not always be the case, with some specimens showing a replacement tooth erupting between two functional teeth (e.g., SAM-PK-K10468, Fig 17C and 17D).

Developmental stages 1–7 are not observed in the sample, as specimens of such early ontogenetic stages of *Galesaurus* have yet to be discovered. However, the smallest known specimen (FMNH PR 1774) approximates the condition of developmental stage 8. In addition, three larger specimens can also be matched to subsequent developmental stages, providing some support for the model. Although the model depicts the replacement patterns hypothesized for the maxillary postcanine series, observations in the mandibular series also fit the model.

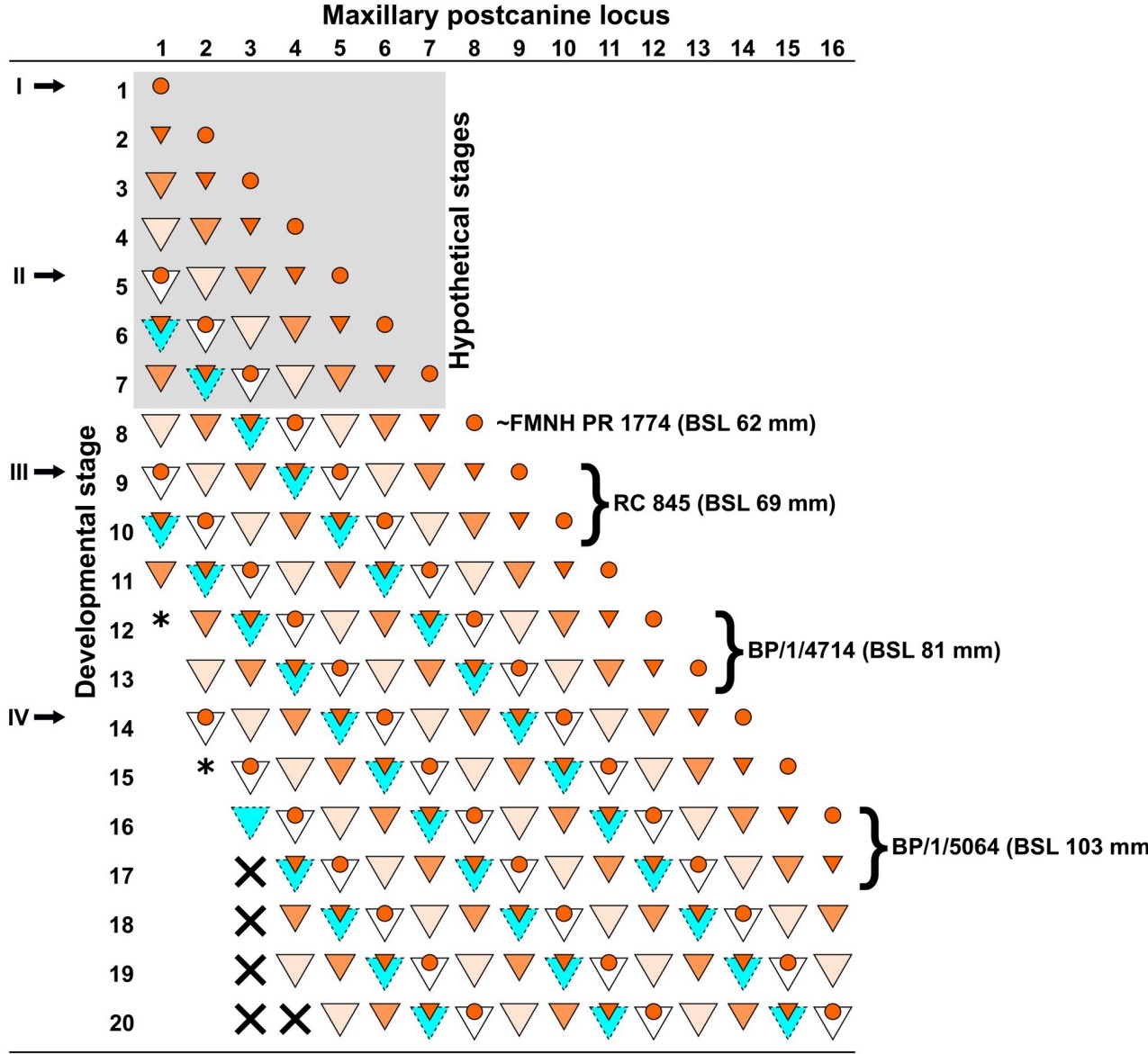

**Fig 23. Hypothetical model of maxillary postcanine replacement in *Galesaurus planiceps* through ontogeny.** Replacement crypts and teeth are indicated by orange circles and small orange triangles respectively. Functional teeth are indicated by large triangles, with shading representing tooth age. Darker tones of orange indicate younger teeth still in the process of developing/erupting, white indicates fully erupted functional teeth, and blue indicates a crown about to be shed due to resorption of the roots. Roman numerals indicate the initiation of a wave of replacement, asterisks (*) indicate the stages at which it is hypothesized that the first active postcanine locus is invaded by the alveolus of the replacement canine, X indicates that a locus has ceased replacement and the alveolus may be filled by bone.

## Paleobiology of *Galesaurus*

Although *Galesaurus* lived contemporaneously with *Thrinaxodon* [4, 8], the different stratigraphic ranges [5] recorded for these two closely related taxa has led to the belief that they occupied different niches [20]. The laterally compressed postcanine teeth of *Galesaurus* have been interpreted as forming a cutting surface [60, 112], and the enlarged, distally directed main cusp of the postcanines might have been an adaptation for capturing and grasping small, wriggling prey [19]. This suggestion that opposing crowns of the maxillary and mandibular

postcanine series created a shearing action is markedly different from the 'piercing' morphology of the dentition of the other basal cynodonts, such as *Procynosuchus* [17]. These assumptions agree with previous proposals that the diet of *Galesaurus* likely consisted of invertebrates and small vertebrates [17, 19, 113]. Furthermore, Abdala et al. [19] suggested that due to the lack of occlusion between opposing postcanines, it is likely that *Galesaurus* did not process food in the oral cavity. Instead, the postcanines may have been used to manipulate prey such that it could be swallowed whole, in a manner comparable to that observed in most extant carnivorous lizards [114].

In *Galesaurus*, the teeth of the postcanine series are arranged in a slight imbricate pattern. Similar overlapping patterns are seen in the maxillary postcanine series of the Late Triassic eucynodont, *Ecteninion lunensis* from Argentina [115], and the mandibular postcanine series of the Early Triassic epicynodont, *Progalesaurus lootsbergensis* from South Africa [7]. A greater degree of overlapping is seen in the distal maxillary postcanines of the tritheledontid, *Riograndia guaibensis* from the Brazilian Norian [116, 117]. The significance of such an arrangement of the postcanines has yet to be determined, however, in *Galesaurus* the canted orientation of the teeth may have allowed for a greater mesial-distal surface area of each tooth crown to contact the prey during jaw adduction. This may have facilitated the swallowing of proportionally large prey items, relative to the body size of *Galesaurus* [19].

The widespread loss of vegetation [6] and the change-over to a *Dicroidium* dominated flora [118], inferred to have been experienced after the end-Permian mass extinction event [119–121], as well as changes in fluvial style (to ephemeral, braided streams) and lowered water tables, are indicative of severe drought conditions during the Early Triassic [17, 122, 123]. This change in environment may have resulted in a decline in the abundance of suitable prey items for *Galesaurus* on which to subsist. In contrast, the generalist dentition of *Thrinaxodon* may have allowed it to survive the grueling climatic changes by becoming an obligate omnivore, the tricuspid and multicuspid dentition of *Thrinaxodon* potentially being better suited for processing plant material than the laterally compressed bicuspid dentition of *Galesaurus*.

Not only are the crown morphologies and replacement patterns of *Galesaurus* different to those of *Thrinaxodon*, but so too is the manner in which the teeth are attached to the jaw. In *Galesaurus* the teeth do not become fused to the bone, instead exhibiting the mammalian condition of permanent gomphosis [60]. In contrast, the teeth of *Thrinaxodon* were shown to ankylose to the bone, in a possible reversal to the basal state within Therapsida [60]. It has been demonstrated that both *Galesaurus* [20, 59] and *Thrinaxodon* [124] grew rapidly for a period of about a year before reaching skeletal maturity. *Thrinaxodon* did so at a smaller body size than has been estimated for *Galesaurus*, suggesting that *Thrinaxodon* may have retained juvenile characters (such as continued replacement of maxillary canines) into adulthood. Alternatively, the hypothesized reversal to the ancestral condition of attachment of the teeth to the jaw via ankylosis, may be coupled with the reversal to the state of continued replacement of the maxillary canine dentition as well.

## Conclusion

This study is the first to apply scanning techniques to determine the changes in dental replacement patterns across an ontogenetic series of *Galesaurus*, and demonstrated that tooth replacement patterns observed in the canines and postcanines of *Galesaurus planiceps* differ from those of *Thrinaxodon liorhinus*. Cessation of the canine replacement occurs in adult *Galesaurus* specimens, i.e., specimens with a BSL larger than ~90 mm, whereas postcanine replacement was recorded in several larger specimens (BSL 94–114 mm). Additionally, the functional canines of adult specimens have an open-rooted morphology, whereas subadults (BSL < 90

mm) have functional canines with closed roots. This change from closed to open-rooted mor-phology of the canines in *Galesaurus* is linked to the cessation of replacement of the canines, and perhaps also to the hypothesized subsequent slowing in the rate of mineralization of the tooth.

*Galesaurus* shows an increase in the number of postcanine teeth in the maxilla and mandi-ble with an increase in skull length, which differs from the condition reported for *Thrinaxo-don*. As in *Thrinaxodon*, the postcanine teeth of *Galesaurus* are replaced in an alternating pattern. However, for *Thrinaxodon* the replacement waves follow each second tooth in the series, whereas in *Galesaurus* the replacement wave affects every third tooth in the series. In both taxa, this pattern has been best observed in the postcanine series of the mandibles.

Although *Thrinaxodon* is widely considered to be a more derived member of the Epicyno-dontia, the evidence for the cessation of replacement in the canines of *Galesaurus* hints that the tooth replacement pattern of *Galesaurus* could be more derived than that of *Thrinaxodon*. In contrast, the apparent retention of the first tooth of the postcanine series in *Galesaurus* is considered a basal condition. *Galesaurus* therefore has a suite of dental characters that are con-currently more basal and more derived than those described in *Thrinaxodon*. Reduction of the number of generations of replacement teeth has previously been described for the postcanine dentition of the gomphodont cynodont *Diademodon*, and the probainognathian *Brasilodon* [125]. The earliest record of the mammalian condition of true diphyodonty occurs in *Morga-nucodon* [29]. The evolutionary pressure behind this adaptation may be as a result of a special-ized diet/foraging behavior. Evidence shows that both *Galesaurus* and *Thrinaxodon* utilized burrows [20–22, 59, 126, 127]. Since *Thrinaxodon* is more prevalent in the fossil record, and has a longer stratigraphic range after the PTB than *Galesaurus*, it may be that specialized dependence on a particular food source led to the earlier extinction of *Galesaurus*, whereas *Thrinaxodon* was able to survive on alternative food sources because of the more generalized morphology of its postcanine crowns.

## Supporting information

**S1 Dataset. Specimens of *Galesaurus planiceps* and *Thrinaxodon liorhinus* included in the study.** These data were used to generate the bivariate plots of the number of postcanine teeth against basal skull length (Figs 21 and 22).
(XLSX)

## Acknowledgments

We thank K. Carlson, T. Jashashvili and K. Jakata for µCT scanning the *Galesaurus* specimens; B. Zipfel and S. Jirah (ESI), S. Kaal and R. Smith (SAM), and E. Butler (NM) for assistance with access to the *Galesaurus* material; and S. Jasinoski, A. LeBlanc and A. Huttenlocker for their helpful comments on a previous version of this manuscript. No permits were required for the described study, which complied with all relevant regulations.

## Author Contributions

**Conceptualization:** Luke A. Norton, Fernando Abdala.

**Formal analysis:** Luke A. Norton.

**Funding acquisition:** Luke A. Norton, Fernando Abdala, Bruce S. Rubidge, Jennifer Botha.

**Investigation:** Luke A. Norton, Fernando Abdala.

**Methodology:** Luke A. Norton, Fernando Abdala.

**Resources:** Fernando Abdala, Bruce S. Rubidge, Jennifer Botha.

**Supervision:** Fernando Abdala, Bruce S. Rubidge, Jennifer Botha.

**Validation:** Fernando Abdala, Bruce S. Rubidge, Jennifer Botha.

**Visualization:** Luke A. Norton.

**Writing – original draft:** Luke A. Norton.

**Writing – review & editing:** Fernando Abdala, Bruce S. Rubidge, Jennifer Botha.

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
