## [Decision Letter · Decision Letter 0]

15 Sep 2020

PONE-D-20-22326

Tooth replacement patterns in the Early Triassic epicynodont *Galesaurus planiceps* (Therapsida, Cynodontia)**

PLOS ONE

Dear Dr. Norton,

Thank you for submitting your manuscript to PLOS ONE. After careful consideration, we feel that it has merit but does not fully meet PLOS ONE’s publication criteria as it currently stands. Therefore, we invite you to submit a revised version of the manuscript that addresses the points raised during the review process.

*==============================*

You have probably benefited from two of the most knowledgeable and qualified scientists to review your manuscript.

It is for this reason that I insist on responding point to point to the minor revision requests they have sent you so that your manuscript benefits from this expertise.

*==============================*

*A rebuttal letter that responds to each point raised by the academic editor and reviewer(s). You should upload this letter as a separate file labeled 'Response to Reviewers'.**A marked-up copy of your manuscript that highlights changes made to the original version. You should upload this as a separate file labeled 'Revised Manuscript with Track Changes'.**An unmarked version of your revised paper without tracked changes. You should upload this as a separate file labeled 'Manuscript'.*

**

We look forward to receiving your revised manuscript.

Kind regards,

Laurent Viriot

Academic Editor

PLOS ONE

*Journal Requirements:*

**

Reviewers' comments:

*Reviewer's Responses to Questions*

***Comments to the Author***

1. Is the manuscript technically sound, and do the data support the conclusions?

*The manuscript must describe a technically sound piece of scientific research with data that supports the conclusions. Experiments must have been conducted rigorously, with appropriate controls, replication, and sample sizes. The conclusions must be drawn appropriately based on the data presented. *

*Reviewer #1: Yes*

*Reviewer #2: Yes*

*2. Has the statistical analysis been performed appropriately and rigorously? *

*Reviewer #1: N/A*

*Reviewer #2: N/A*

*3. Have the authors made all data underlying the findings in their manuscript fully available?*

*The PLOS Data policy requires authors to make all data underlying the findings described in their manuscript fully available without restriction, with rare exception (please refer to the Data Availability Statement in the manuscript PDF file). The data should be provided as part of the manuscript or its supporting information, or deposited to a public repository. For example, in addition to summary statistics, the data points behind means, medians and variance measures should be available. If there are restrictions on publicly sharing data—e.g. participant privacy or use of data from a third party—those must be specified.*

*Reviewer #1: Yes*

*Reviewer #2: Yes*

*4. Is the manuscript presented in an intelligible fashion and written in standard English?*

*PLOS ONE does not copyedit accepted manuscripts, so the language in submitted articles must be clear, correct, and unambiguous. Any typographical or grammatical errors should be corrected at revision, so please note any specific errors here.*

*Reviewer #1: Yes*

*Reviewer #2: Yes*

*5. Review Comments to the Author*

*Please use the space provided to explain your answers to the questions above. You may also include additional comments for the author, including concerns about dual publication, research ethics, or publication ethics. (Please upload your review as an attachment if it exceeds 20,000 characters)*

*Reviewer #1: I have very few substantive reviewer comments for this manuscript. Most of my edits/suggestions are minor organizational issues that I have outlined in the attached PDF.*

My one non-organizational comment would be to make sure the authors include virtual CT slices of all of the specimens for which CT data were available. I understand there are good reasons not to include the full CT datasets for this manuscript, but make sure that enough CT slices are included to corroborate all of the interpretations made in this paper. I think most if not all of them are here, but the more the better. Any additional, good-quality CT images of the dentine fragments from the incisors/canines/postcanines would be worthwhile to include either as in-text figures or as supplementary data.

Aside from that and my minor in-text comments, I thought the paper was well-written, fairly clear and concise, and provided worthwhile interpretations of the data. I recommend a minor change to one of the last figures as well.

Sincerely,

*Aaron LeBlanc*

*Reviewer #2: Review of “Tooth replacement patterns in the Early Triassic epicynodont Galesaurus*

planiceps (Therapsida, Cynodontia)” Manuscript for PONEW by Norton, L. A., Abdala, F. Rubidge B. S. and J. Botha

By: Zhe-Xi Luo (zxluo@uchicago.edu)

Review Summary

This is a significant study on the tooth replacement pattern of the cynodont Galesaurus planiceps, with broad implications for the evolution of tooth replacement and skull growth in cynodont evolution. Because this cynodont is a Triassic relative of mammals, its anatomical structure, especially the skull growth and tooth replacement patterns, will be very important for interpreting the evolution from cynodonts to mammals.

The characterization of the tooth replacement is detailed, informative, and excellent. The CT visualization is also very beautiful and very appealing. It adds significant new information on the wider evolutionary variation of the tooth morphology and replacement pattern for epicynodonts (a clade include mammals).

This is also a thorough paper, and well done. I agree with the key conclusions by the authors, and strongly support for Plos-One to publish this paper.

Suggestions for Revision

The paper is publishable – in a very good shape already. I have several suggestions for optional revision. But none of these is about the substance. I offer these comments, in the spirit of making the paper even better, and to make the paper a more useful key reference for other paleontologists, like myself.

Suggestion 1

P2 Lines 33-41 – Abstract

The several sentences of the abstract (lines 33-41) should be re-arranged slightly so the delivery would be more orderly. I suggest these changes of the abstract

How about change to “….. The growth series of Galesaurus shows that the incisors and canine continue to develop and replace, even in the largest (presumably the oldest) specimens of Galesaurus. In Galesaurus, replacement of the canine ceased with the attainment of skeletal maturity, whereas in Thrinaxodon, replacement of the canines continued into adulthood. In the maxilla, the first postcanine is consistently the smallest tooth, showing a proportional reduction in size as skull length increased. The first postcanine locus underwent longer development and sustained its replacement in Galesaurus than in Thrinaxodon. A longer retention of this tooth is a key difference of Galesaurus from Thrinaxodon in which the anterior-most postcanines are lost after replacements. This difference have contributed to the lengthening of the postcanine series as teeth continue to be added to the posterior end of toothrow through ontogeny in Galesaurus, which differs from Thrinaxodon. Overall, there are considerable differences between Galesaurus and Thrinaxodon relating to the replacement and development of their teeth. “

Suggestion 2

The paper will be better if it can add an illustration of CT visualization of the in situ teeth in the mandible in the medial view of the jaw. It is important to illustrate this view, easily obtainable from the CT scans. Several dental replacement features can be shown.

In Thrinaxodon the dental lamina groove (Crompton’s groove) is present on along the margin of the dental alveoli, in the medial view of the mandible. The authors have clearly explained that the tooth alveoli are not ankylosed in Galesaurus, and the gomphodont tooth implant in the alveoli are permanent. By comparison, in Thrinaxodon the deciduous tooth become ankylosed in the dentary (alveoli would be ossified) before the tooth was shed and lost. The dental lamina groove is related to this pattern in Thrinaxodon. Does Galesaurus have the dental lamina groove (for replacement). If dental lamina groove is absent, then there is one more difference from Thrinaxodon.

I suggest that the authors can add this description/discussion with a figure on Galesaurus. This will make the paper better by providing more comparative information to Thrinaxodon.

Suggestion 3

There are now several CT scan studies on Galesaurus skulls (Jasinoski and Abdala 2017; Pusch et al. 2019, and the present manuscript by Norton et al.). But these otherwise excellent papers have not shown a good CT visualization of the postcanine tooth crown. Tooth crowns of Galesaurus are the most unique, among cynodonts , and are quite different from those of Thrinaxodon. Can you illustrate an upper PC crown in medial view, and lateral view. And then do the same for a lower postcanine? The line drawings in Figure 1 are just too simple, and do not show the lingual views.

This will make the paper much more useful, for other paleontologists.

Suggestion 4 (also a neutral comment)

The upper canine can be open-rooted, even in larger and presumably the older skulls (e.g., NMQR135; BP/1/5064). This is most interesting. This could be due to the fact the canine was still slowly growing, or that the tooth is high-crown, or deep rooted, or both.

Interestingly, there is an analogous pattern in the Jurassic haramiyidan Vilevolodon. The replacing lower incisor can also be open-rooted, in the incisor locus that experienced slow tooth growth and replacement in some of these haramiyidans (Luo et al. 2017: extended data figures 4 and 6). I think the open-rooted pattern of incisors in the haramiyidan Vilevolodon may be convergent in the tooth growth mode to the open-rooted and high-crown canines (but obviously these incisor and canine are in different tooth positions).

This is consistent with the authors’ interpretation that Galesaurus has a relatively slow, and sustained tooth replacement.

*See: Luo, Z.-X., Meng, Q.-J., Grossnickle, D. M., Liu, D., Zhang, Y.-G., Neander, A. I., and Q, Ji. 2017. New evidence for mammaliaform ear evolution and feeding adaptation in a Jurassic ecosystem. Nature. 548: 326-329 (doi:10.1038/nature23483).*

*6. PLOS authors have the option to publish the peer review history of their article (what does this mean?). If published, this will include your full peer review and any attached files.*

**

*Reviewer #1: **Yes: **Aaron LeBlanc*

*Reviewer #2: **Yes: **Zhe-Xi Luo*

**

*While revising your submission, please upload your figure files to the Preflight Analysis and Conversion Engine (PACE) digital diagnostic tool, https://pacev2.apexcovantage.com/. PACE helps ensure that figures meet PLOS requirements. To use PACE, you must first register as a user. Registration is free. Then, login and navigate to the UPLOAD tab, where you will find detailed instructions on how to use the tool. If you encounter any issues or have any questions when using PACE, please email PLOS at figures@plos.org. Please note that Supporting Information files do not need this step.*

---

## [Author Response · Author response to Decision Letter 0]

4 Nov 2020

Reviewer 1

I have very few substantive reviewer comments for this manuscript. Most of my edits/suggestions are minor organizational issues that I have outlined in the attached PDF.

—All corrections suggested in the marked-up PDF were incorporated into the manuscript. A detailed list of the changes made are listed below:

• Page 4, Line 82: Maybe add a brief paragraph on why studying these tooth replacement patterns is important (I'm assuming this would relate to the transition from "reptilian-style" polyphyodonty to oligophyodonty).—Moved sentence from p. 6 as suggested, and modified to read “Determining the tooth replacement patterns in Galesaurus and comparing them to those observed in the previously studied cynodont Thrinaxodon [44] may help to ascertain whether any variation in tooth replacement pattern occurred amongst the basal-most cynodonts.”

• Page 6, Lines 117–120: This should probably be included on page 4 where I included my comment.—See comment above.

• Page 6, Lines 137–140. Novelty of study clearly stated.—No change necessary

• Page 7, Line 148: Inserted “(e.g., tooth replacement)”

• Page 10, Line 200: Inserted “of tooth development”

• Page 12, Line 240: Do you have a figure number to refer to here?―Citations to two new figures (Figs 3 and 5) added, as well as a citation to Fig 4 (fig 5 of original submission). The relevant captions were inserted or moved as necessary.

• Page 14, Line 279: Inserted “...and were still in the process of developing”

• Page 14, Line 283: Move this interpretation of open roots up to the first time you mention open roots in the maxillary incisors.—Moved “and was still in the process of developing.” Changed “In contrast” to “whereas” 

• Page 14, Line 284: And closed/tapering at the root apex? How do the roots of developing vs. fully developed teeth differ?—Inserted “…and tapers to a closed point at the root apex” at the end of the sentence.

• Page 15, Lines 308–309: Inserted “is”

• Page 31, Line 798: Inserted “The”

• Page 33, Lines 739–740: Please reference your graphs here (e.g., Fig. 16).—Inserted “(e.g., Figs 21 and 22)” and moved captions accordingly. These were figs 16 and 17 of the original submission.

• Page 34, Lines 763–767: A bit unclear: the previous paragraph and the start of this sentence led me to believe that the PC crowns become more complex in larger individuals, but the end of this statement suggests the opposite. Are you suggesting that the accessory cusps of the PC's are lost through successive replacement events in Galesaurus? Could clarify that a bit here.—Deleted “This condition is similar to that described for Diademodon [49] and Nanictosaurus [84], where the distal elements of the postcanine series have a more complex crown morphology. This suggests that the postcanine crown morphologies become more simplified with each successive replacement.” (see also comment on P. 35, L. 806–808)

• Page 35, Line 797: Inserted “the”

• Page 35, Line 798: So this would be a similar trend to that observed in Galesaurus? Could make a comparison to your findings here.—Inserted “This condition is similar to that observed in Galesaurus.”

• Page 35, Lines 806–808: This is repetition of the discussion on pg. 34. Please combine these together, or even easier: delete part of the discussion on pg. 34 and leave this one. No need to keep repeating the main findings.—See comment for P 34, L. 763–767 above. Inserted “and Nanictosaurus [84].”

• Page 36, Lines 814–819: Any inferences to be made about this imbrication? Does it relate to diet in some way? I suggest moving this to the results otherwise.—This paragraph was moved to the section “Paleobiology of Galesaurus”. Inserted “The significance of such an arrangement of the postcanines has yet to be determined, however, in Galesaurus the canted orientation of the teeth may have allowed for a greater mesial-distal surface area of each tooth crown to contact the prey during jaw adduction. This may have facilitated the swallowing of proportionally large prey items, relative to the body size of Galesaurus [19].”

• Page 36, Line 828: In Galesaurus or is this a broader statement? If this is meant to be a broader statement, I would add two qualifiers: MARGINAL replacement teeth are always positioned lingual (palatal teeth frequently do not follow this "rule"). And I would also cite Edmund, 1960. If this is a specific reference to Galesaurus, I suggest clarifying that here.—Inserted “In Galesaurus,”

• Page 37, Lines 836–837: Can you speculate on the direction of the replacement wave? For example, when the odd-numbered teeth are at a different replacement stage, which of them, I1 or I3, is at a more advanced stage?—Inserted the following paragraph: “It was not possible to determine which direction the replacement wave travelled from the present sample. However, in specimens that show evidence of two or more replacement incisors in a premaxilla (e.g., RC 845, BP/1/5064 and SAM-PK-K10468), the distal teeth of the odd and even-numbered waves tend to be larger than the mesial teeth (i.e., I3 larger than I1, and I4 large than I2). This suggests a back-to-front movement of the replacement wave, but we cannot rule out the idea that the mesial tooth may be smaller, because it represents the initiation of the next replacement wave. If this is correct, then it would support the replacement waves moving in a front-to-back direction.”

• Page 38, Lines 869–873: Good comparisons here.—No change necessary

• Page 42, Line 972: Replaced “simple” with “easy”

• Page 43, Line 1007: Inserted “of the entire postcanine series”

• Page 43, Line 1008: Clarify what you mean by "cycle" in my previous comment in this sentence and this should be more clear.—See comment above. Inserted “replacement”

• Page 45, Line 1044: This does not necessarily have to be its own section. Fig. 18 is essentially a summary of what was built upon in the previous section.—Deleted the heading “Postcanine replacement model”

• Page 45, Line 1046: Replaced “is not perfect for the following reasons” with “has two limitations”

• Page 47, Line 1099: Inserted “within Therapsida”

• Page 47, Line 1106: Replaced “too” with “as well”

• Page 48, Lines 1129–1131: Definitely an appropriate conclusion, given the data and comparisons. What about the postcanine replacement patterns/frequency?—Inserted “In contrast, the apparent retention of the first tooth of the postcanine series in Galesaurus is considered a basal condition. Galesaurus therefore has a suite of dental characters that are concurrently more basal and more derived than those described in Thrinaxodon.”

• Fig 18: Add a label for stages 1–7 indicating that these are hypothetical, just for clarification. This is important, because it's unlikely that tooth initiation started at locus 1 in the PC series. There are plenty of papers discussing tooth initiation patterns in vertebrates, which point to all kinds of ways that a tooth series may initiate.—See detailed response below

My one non-organizational comment would be to make sure the authors include virtual CT slices of all of the specimens for which CT data were available. I understand there are good reasons not to include the full CT datasets for this manuscript, but make sure that enough CT slices are included to corroborate all of the interpretations made in this paper. I think most if not all of them are here, but the more the better. Any additional, good-quality CT images of the dentine fragments from the incisors/canines/postcanines would be worthwhile to include either as in-text figures or as supplementary data.

—We have included several additional figures showing virtual transverse sections through the anterior and postcanine dentitions of RC 845 (Figs 3 and 7) and BP/1/4602 (Figs 5 and 10).

Aside from that and my minor in-text comments, I thought the paper was well-written, fairly clear and concise, and provided worthwhile interpretations of the data. I recommend a minor change to one of the last figures as well.

―We have added a clarification to Fig 23 (fig 18 of original submission) that Developmental Stages 1–7 represent hypothetical stages not preserved in the currently known specimens of Galesaurus.

Reviewer 2

Suggestion 1

The several sentences of the abstract (lines 33-41) should be re-arranged slightly so the delivery would be more orderly. I suggest these changes of the abstract

How about change to “…The growth series of Galesaurus shows that the incisors and canine continue to develop and replace, even in the largest (presumably the oldest) specimens of Galesaurus. In Galesaurus, replacement of the canine ceased with the attainment of skeletal maturity, whereas in Thrinaxodon, replacement of the canines continued into adulthood. In the maxilla, the first postcanine is consistently the smallest tooth, showing a proportional reduction in size as skull length increased. The first postcanine locus underwent longer development and sustained its replacement in Galesaurus than in Thrinaxodon. A longer retention of this tooth is a key difference of Galesaurus from Thrinaxodon in which the anterior-most postcanines are lost after replacements. This difference have contributed to the lengthening of the postcanine series as teeth continue to be added to the posterior end of toothrow through ontogeny in Galesaurus, which differs from Thrinaxodon. Overall, there are considerable differences between Galesaurus and Thrinaxodon relating to the replacement and development of their teeth.

―The abstract was modified following the reviewer’s suggestions.

Suggestion 2

The paper will be better if it can add an illustration of CT visualization of the in situ teeth in the mandible in the medial view of the jaw. It is important to illustrate this view, easily obtainable from the CT scans. Several dental replacement features can be shown.

In Thrinaxodon the dental lamina groove (Crompton’s groove) is present on along the margin of the dental alveoli, in the medial view of the mandible. The authors have clearly explained that the tooth alveoli are not ankylosed in Galesaurus, and the gomphodont tooth implant in the alveoli are permanent. By comparison, in Thrinaxodon the deciduous tooth become ankylosed in the dentary (alveoli would be ossified) before the tooth was shed and lost. The dental lamina groove is related to this pattern in Thrinaxodon. Does Galesaurus have the dental lamina groove (for replacement). If dental lamina groove is absent, then there is one more difference from Thrinaxodon.

I suggest that the authors can add this description/discussion with a figure on Galesaurus. This will make the paper better by providing more comparative information to Thrinaxodon.

―The medial view of the right mandible of AMNH FARB 2227 was illustrated by (Pusch et al., 2019: fig. 16B), however the authors did not comment on the presence of a groove to house the dental lamina. The dental lamina groove is visible in the virtual horizontal sections through the postcanine series of adult specimens (Figs 15 and 17 [figs 10 and 12 of first submission]). The label “dl” has been added to the figures, and the figure captions updated to include the abbreviation “dl, dental lamina groove.” Additionally, Rigney (1938) described a canal lingual to the postcanine series, and the following sentences were inserted on P. 42: “Furthermore, Rigney [63] described two canals in the dentary lingual to the postcanine series. The description of these canals is similar to that of the longitudinal grooves described in the maxilla and dentary of Thrinaxodon [40]. Crompton [40] proposed that these grooves housed the dental lamina. Similar lingual grooves extending the length of the postcanine series are evident in two adult specimens of Galesaurus (BP/1/5064 and SAM-PK-K10468, Figs 15 and 17).”

Suggestion 3

There are now several CT scan studies on Galesaurus skulls (Jasinoski and Abdala 2017; Pusch et al. 2019, and the present manuscript by Norton et al.). But these otherwise excellent papers have not shown a good CT visualization of the postcanine tooth crown. Tooth crowns of Galesaurus are the most unique, among cynodonts , and are quite different from those of Thrinaxodon. Can you illustrated an upper PC crown in medial view, and lateral view. And then do the same for a lower postcanine? The line drawings in Figure 1 are just to simple, and do not show the lingual views.

This will make the paper much more useful, for other paleontologists.

―A new figure (Fig 6) showing the labial and lingual crown morphology of two upper and two lower postcanine teeth was added.

Suggestion 4 (also a neutral comment)

The upper canine can be open-rooted, even in larger and presumably the older skulls (e.g., NMQR135; BP/1/5064). This is most interesting. This could be due to the fact the canine was still slowly growing, or that the tooth is high-crown, or deep rooted, or both.

Interestingly, there is an analogous pattern in the Jurassic haramiyidan Vilevolodon. The replacing lower incisor can also be open-rooted, in the incisor locus that experienced slow tooth growth and replacement in some of these haramiyidans (Luo et al. 2017: extended data figures 4 and 6). I think the open-rooted pattern of incisors in the haramiyidan Vilevolodon may be convergent in the tooth growth mode to the open-rooted and high-crown canines (but obviously these incisor and canine are in different tooth positions).

This is consistent with the authors’ interpretation that Galesaurus has a relatively slow, and sustained tooth replacement.

―Although we agree that the roots of the replacement incisors of Vilevolodon are open-rooted, we disagree with comparing them to the maxillary canines of the larger specimens of Galesaurus (e.g., BP/1/5064 and NMQR 135). The main reason, in these two Galesaurus specimens the open-rooted tooth is already functional, whereas in Vilevolodon the open-rooted tooth is clearly a developing replacement tooth with the functional tooth still in situ. It would perhaps be better to compare the condition in the mandibular incisors of Vilevolodon to those of the maxillary canines in the smaller specimens of Galesaurus (e.g., RC 845 and BP/1/4602), or perhaps even better, the left mandibular canine of BP/1/4602. In these examples, the replacing tooth may be considered as open-rooted as they were still in the process of developing, and the corresponding functional teeth have yet to be exfoliated.

---

## [Decision Letter · Decision Letter 1]

2 Dec 2020

Tooth replacement patterns in the Early Triassic epicynodont *Galesaurus planiceps* (Therapsida, Cynodontia)

PONE-D-20-22326R1

Dear Dr. Norton,

We’re pleased to inform you that your manuscript has been judged scientifically suitable for publication and will be formally accepted for publication once it meets all outstanding technical requirements.

Kind regards,

Laurent Viriot

Academic Editor

PLOS ONE

Additional Editor Comments (optional):

Reviewers' comments:

Reviewer's Responses to Questions

**Comments to the Author**

1. If the authors have adequately addressed your comments raised in a previous round of review and you feel that this manuscript is now acceptable for publication, you may indicate that here to bypass the “Comments to the Author” section, enter your conflict of interest statement in the “Confidential to Editor” section, and submit your "Accept" recommendation.

Reviewer #1: All comments have been addressed

Reviewer #2: All comments have been addressed

2. Is the manuscript technically sound, and do the data support the conclusions?

Reviewer #1: Yes

Reviewer #2: Yes

3. Has the statistical analysis been performed appropriately and rigorously? 

Reviewer #1: N/A

Reviewer #2: N/A

4. Have the authors made all data underlying the findings in their manuscript fully available?

Reviewer #1: Yes

Reviewer #2: Yes

5. Is the manuscript presented in an intelligible fashion and written in standard English?

Reviewer #1: Yes

Reviewer #2: Yes

6. Review Comments to the Author

Reviewer #1: (No Response)

Reviewer #2: (No Response)

7. PLOS authors have the option to publish the peer review history of their article (what does this mean?). If published, this will include your full peer review and any attached files.

Reviewer #1: **Yes: **Aaron LeBlanc

Reviewer #2: No

---

## [Editor Report · Acceptance letter]

7 Dec 2020

PONE-D-20-22326R1 

Tooth replacement patterns in the Early Triassic epicynodont *Galesaurus planiceps* (Therapsida, Cynodontia) 

Dear Dr. Norton:

I'm pleased to inform you that your manuscript has been deemed suitable for publication in PLOS ONE. Congratulations! Your manuscript is now with our production department. 

Kind regards, 

on behalf of

Dr. Laurent Viriot 

Academic Editor

PLOS ONE